**CYCLOPs: A Unified Framework for Surface Flux-Driven Cyclones Outside the Tropics**
Kerry Emanuel (1), Tommaso Alberti (2), Stella Bourdin (3), Suzana J. Camargo (4,5), Davide
Faranda (6,7,8), Emmanouil Flaounas (9,10), Juan Jesus Gonzalez-Aleman (11), Chia-Ying Lee (4),
Mario Marcello Miglietta (12), Claudia Pasquero (13), Alice Portal (14), Hamish Ramsay (15), Marco
Reale (16), Romualdo Romero (17)
(1)  Massachusetts Institute of Technology, 77 Mass. Ave., Cambridge, MA 02139
(2) Istituto Nazionale di Geofisica e Vulcanologia, Rome, Italy
(3) Atmospheric, Oceanic and Planetary Physics, Department of Physics, University of Oxford,
9          Oxford, United Kingdom

(4) Lamont-Doherty Earth Observatory, Columbia University, Palisades, New York, USA
(5) Columbia Climate School, Columbia University, New York, New York, USA
(6) Laboratoire des Sciences du Climat et de l'Environnement, UMR 8212 CEA-CNRS-UVSQ,
13          Université Paris-Saclay & IPSL, CE Saclay l'Orme des Merisiers, 91191 Gif-sur-Yvette,
14          France

(7) London Mathematical Laboratory, 8 Margravine Gardens, London W6 8RH, UK
(8) LMD/IPSL, ENS, Université PSL, École Polytechnique, Institut Polytechnique de Paris, Sorbonne
17          Université, CNRS, Paris France

(9) Institute for Atmospheric and Climate Science, ETH Zurich, Zurich, Switzerland
(10) Institute of Oceanography, Hellenic Centre for Marine Research, Athens, Greece
(11) Spanish State Meteorological Agency, AEMET, Department Development and Applications
(12) CNR-ISAC, Padua, Italy
(13) Department of Earth and Environmental Sciences, University of Milano - Bicocca, Italy
(14) Institute of Atmospheric Sciences and Climate (CNR-ISAC), National Research Council of Italy,
24          Bologna, Italy

(15) CSIRO Environment, Aspendale, Victoria, Australia
(16) National Institute of Oceanography and Applied Geophysics - OGS, Trieste, Italy
(17) Grup de Meteorologia, Departament de Física, Universitat de les Illes Balears, Palma de
28          Mallorca, Spain

Correspondence: Kerry Emanuel (emanuel@mit.edu)

30                                                    Abstract

Cyclonic storms resembling tropical cyclones are sometimes observed well outside the
tropics. These include medicanes, polar lows, subtropical cyclones, Kona storms, and
possibly some cases of Australian East Coast Lows. Their structural similarity to tropical
cyclones lies in their tight, nearly axisymmetric inner cores, eyes, and spiral bands.
Previous studies of these phenomena suggest that they are partly and sometimes
wholly driven by surface enthalpy fluxes, as with tropical cyclones. Here we show,
through a series of case studies, that many of these non-tropical cyclones have
morphologies and structures that resemble each other and also closely match those of
tropical transitioning cyclones, with the important distinction that the potential intensity
that supports them is not present in the pre-storm environment but rather is locally
generated in the course of their development. We therefore propose to call these storms
CYClones from Locally Originating Potential intensity (CYCLOPs). We emphasize that
mature CYCLOPs are essentially the same as tropical cyclones but their development
requires substantial modification of their thermodynamic environment on short time
scales. Like their tropical cousins, the rapid development and strong winds of CYCLOPs
pose a significant threat and forecast challenge for islands and coastal regions, and the
effects of climate change on them should be considered.

## 1.    Introduction

Cyclones that resemble tropical cyclones are occasionally observed to develop well outside the
tropics. These include polar lows, medicanes, subtropical cyclones, Kona storms (central North
Pacific), and perhaps some cases of Australian East Coast Lows. The identification of such
systems is usually based on their appearance in satellite imagery and on the environmental
conditions in which they occur. Here we show that many of these systems are manifestations of
the same physical phenomenon and, as such, should be given a common, physically-based
designation. We propose to call these CYClones from Locally Originating Potential intensity
(CYCLOPs), with reference to the one-eyed creatures of Greek mythology[1]. We show that in
many respects these developments resemble classical "tropical transition" (TT) events (e.g.
Bosart and Bartlo, 1991), but they are distinguished from the latter by occurring in regions
where the climatological potential intensity is small or zero. (Some events previously identified
as TT events were probably examples of CYCLOPs.) Their often-rapid development and
intense mesoscale inner cores, compared to extratropical cyclones, make CYCLOPs significant
hazards and a forecasting challenge.

---

[1] Late in the process of writing this paper, we discovered that there is a scientific research project by the
same name, standing for "Improving Mediterranean CYCLOnes Predictions in Seasonal forecasts with
artificial intelligence" (https://www.cmcc.it/projects/CYCLOPs-improving-mediterranean-cyclones-
predictions-in-seasonal-forecasts-with-artificial-intelligence). Its team leader, Leone Cavicchia, has
graciously agreed with our use of the same name.

Cyclones of synoptic and sub-synoptic scale are powered by one or both of two energy sources:
the available potential energy (APE) associated with isobaric temperature gradients
(baroclinity), and fluxes of enthalpy (sensible and latent heat) from the surface (usually the
ocean) to the atmosphere[2]. A normal extratropical cyclone over land is an example of the
former, while the latter is epitomized by a classical tropical cyclone. Extratropical transitioning
and tropical transitioning cyclones[3] can be  powered by both sources, either in sequence or with
the relative proportion varying over the life of the storm (see, e.g., Fantini, 1990). Additionally,
we note that tropical cyclones often originate in disturbances, such as African easterly waves,
that derive their energy from baroclinic and barotropic sources.
Like tropical cyclones, CYCLOPs are mainly powered by surface enthalpy fluxes, but differ from
the former in that the required potential intensity is produced locally and transiently, whereas
tropical cyclones develop in seasons and regions where sufficient potential intensity is always
present. CYCLOPs closely resemble the strongly baroclinic cases of tropical transition defined
and discussed by Davis and Bosart (2004), except that they occur in regions where the
climatological potential intensity is too small for tropical cyclogenesis, relying on synoptic-scale
perturbations that locally enhance potential intensity in space and time. Davis and Bosart (2004)
confined their attention to tropical cyclone formation in regions of high sea surface temperature,
stating that "The precursor cyclone must occlude and remain over warm water ($\gtrsim$26°C) for at
least a day following occlusion." Similarly, McTaggart-Cowan et al. (2008) and McTaggart-
Cowan et al. (2013) only examined cases of tropical transition that resulted in named tropical
cyclones. But McTaggert-Cowan et al. (2015) recognized that around 5% of the cases they
identified as tropical transition cases occurred over colder water and that upper-level troughs
played a key role in destabilizing the atmosphere with respect to the sea surface. We here build
on this work and place it within the framework of potential intensity theory.
The basic physics of CYCLOPs was explored by the first author in reference to medicanes
(Emanuel, 2005), and is illustrated in Figure 1. While the actual evolution is, of course,
continuous, it is simpler to discuss it in phases. For this narrow purpose, we assume that a cut-
off cyclone has already formed in the upper troposphere, signified by the presence of an
isolated potential vorticity (PV) anomaly near the tropopause (Figure 1a). We show an idealized
circularly symmetric anomaly and a cross-section through it; in practice, such developments are

---

[2] Some would regard latent heating as an additional energy source, but condensation through a deep layer is present in most cyclones, being strongly tied to vertical motion, and one could argue that it should not be regarded as an external heat source but rather as a modification of the static stability (Emanuel et al., 1994). On the other hand, large areas of cyclones are not water saturated and various processes affect the availability of moisture in such regions, and thus, ultimately, the amount of latent heat release (e.g. Winschall et al., 2014). Moreover, latent heat release, when it is strong enough, can lead to the phenomenon of diabatic Rossby waves (Boettcher and Wernli, 2013; Kohl and O'Gorman, 2022; Parker and Thorpe, 1995; Wernli et al., 2002) and here the partitioning between advection and latent heating is important.

[3] Extratropical transitioning cyclones are storms whose energy source is transitioning from surface fluxes to ambient baroclinity, while the energy source of tropical transitioning cyclones is moving in the opposite direction.

seldom close to axisymmetric in their earlier stages. In the illustration, the first phase is
assumed to occur over land, but that need not be the case in general.

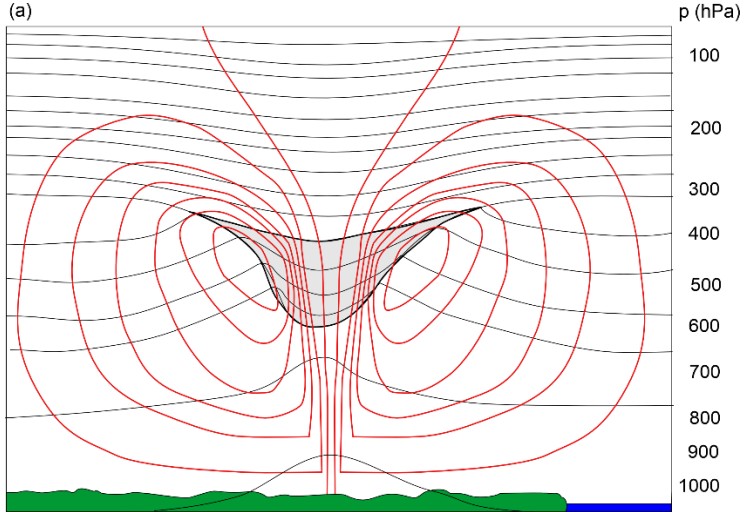

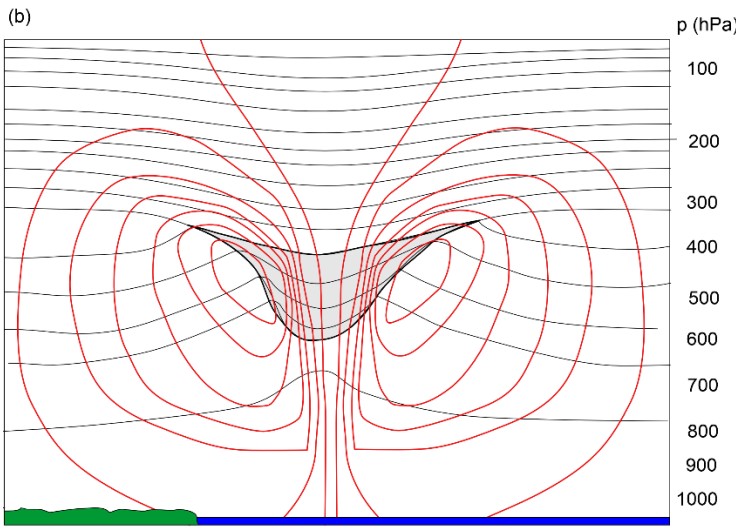

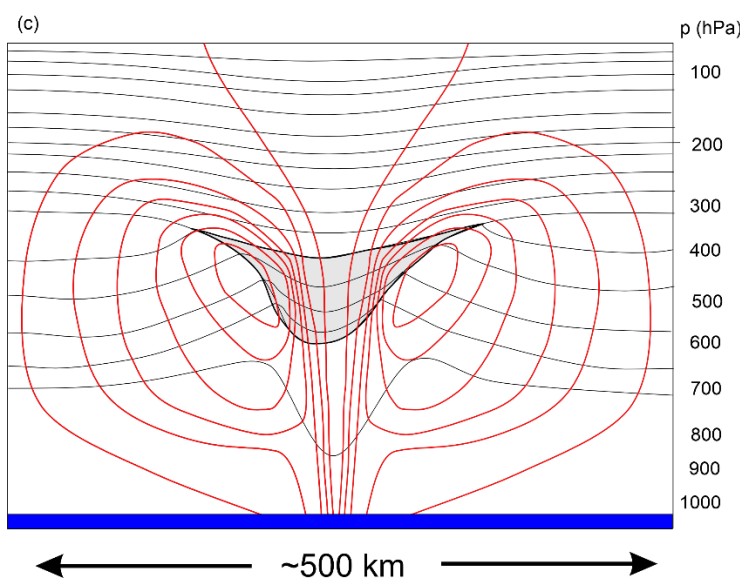

*Figure 1: Three stages in the development of a CYCLOP. Each of the three panels shows a cross-section through an isolated PV anomaly at the tropopause (gray shading), idealized as circular. The thin black curves are isentropes, while the red curves are isotachs of the flow normal to the cross-section. In the first phase (a), the PV anomaly is over land, the troposphere beneath it is anomalously cold, and there may be no flow at the surface. The system moves out over open water in the next phase (b) and warming of the boundary layer and lower troposphere suffices to eliminate the cold anomaly at and near the surface. Weak cyclonic flow develops in response to the positive PV anomaly aloft. In the final phase (c), the surface heat fluxes become important and a tight, inner warm core develops.*

During the creation of the near-tropopause PV anomaly, some combination of lifting and cold air
seclusion has cooled and humidified the column underneath the PV anomaly, and the cold
anomaly extends right to the surface. From a PV inversion perspective, the near-surface
anticyclone that results from inverting the negative potential temperature anomaly at the surface
is assumed to just cancel the cyclonic anomaly that results from inverting the tropopause PV
anomaly, yielding no circulation at the surface. This assumption is made for simplicity here and
we do not mean to imply that such a cancellation is common.
In phase 2 (Figure 1b), the system drifts out over open water that is warm enough to diminish
and eventually eliminate the cold anomaly at the surface, "unshielding" it from the PV anomaly
aloft. During this phase, a cyclonic circulation develops in the lower troposphere, with a
horizontal scale commensurate with that of the PV anomaly.
If the local potential intensity is large enough, and the air aloft sufficiently close to saturation,
Wind-Induced Surface Heat Exchange (WISHE) can develop a tropical cyclone-like vortex
(phase 3, Figure 1c), with a warm inner core, eye and eyewall, and perhaps spiral bands. Note
that the inner core may be warm only with respect to the synoptic-scale cold anomaly
surrounding it, not necessarily with respect to the distant environment. At mid-levels, the
temperature anomaly may manifest as a small-scale warm anomaly surrounded by a synoptic-
scale cold anomaly.
In reality, these phases blend together into a continuum. Moreover, all three phases may (and
often do) occur over water. One practical challenge is calculating the potential intensity. This
should be calculated using the temperatures of the sea surface and the free troposphere under
the PV anomaly aloft, but before the troposphere has appreciably warmed from surface fluxes.
In practice, because the warming occurs either as the PV anomaly develops over water, or as it
moves over water from land, we have no access to the sounding of the free troposphere *before*
it has warmed up. The true potential intensity for a TC developing within the cold column under
the PV anomaly is hence impossible to obtain. Nevertheless, we can estimate how cold the
troposphere was before surface fluxes warmed it by using the surface pressure perturbation as
a proxy for the CYCLOPs-induced warming, and assuming that the troposphere has an
approximately moist adiabatic temperature profile[4].  This is derived in the Appendix. The result
is a modified potential intensity, $V_{pm}$, given by
$$V_{pm}^2 = V_p^2 - \frac{C_k}{C_D} \frac{T_s}{T_{400}} \phi'_{950},$$
( 1)

where $V_p$ is the potential intensity calculated in the usual way (e.g. Bister and Emanuel, 2002)
from the local sea surface temperature and atmospheric sounding, $\phi'_{950}$ is the perturbation away
from climatology of the near-surface geopotential, which we here evaluate at 950 hPa, $C_k$ and

---

[4] In regions of deep convection, lapse rates are usually observed to be close to moist adiabatic (Betts, 1986)

$C_D$ are the surface exchange coefficients for enthalpy and drag, and $T_s$ and $T_{400}$ are the
absolute temperatures at the surface and 400 hPa. As explained in the Appendix, we
approximate the coefficient multiplying $\phi'_{950}$ in (1) by a constant value, 1.3, a reasonable estimate
of the mean value of the coefficient. Note that (1) is only valid in places where our assumption
that the lapse rate is moist adiabatic trough a deep layer is valid and deep moist convection
occurs. Also, in the early stages of development, the outflow level is likely to be lower and the
outflow temperature correspondingly higher, reducing the value of the coefficient multiplying $\phi'_{950}$
in (1).
Beginning with an upper cold cyclone in an environment of otherwise zero potential intensity,
Emanuel (2005) simulated the development of a WISHE-driven cyclone using the axisymmetric,
nonhydrostatic hurricane model of Rotunno and Emanuel (1987). The cold, moist troposphere
under such an upper cyclone proves to be an ideal incubator of surface flux-driven cyclones
with characteristics nearly identical to those of tropical cyclones. One interesting facet of the
process is that the anomalous surface enthalpy flux destroys the parent upper cold low over a
period of a few days, through the action of deep convection. This in another way in which
CYCLOPs are distinguished from tropical cyclones, which tap spatially extensive regions of high
potential intensity.
## 2.    Why it matters
Why should we care whether a cyclone is driven by surface fluxes or baroclinity? From a
practical forecasting standpoint, the spatial distributions of weather hazards, like rain and wind,
can be very different, as can be the development time scales and fundamental predictability.
In classical baroclinic cyclones, the strongest winds are often found in frontal zones and can be
far from the cyclone center, while precipitation is usually heaviest in these frontal zones and in a
shield of slantwise ascent extending poleward from the surface cyclone. There is long
experience in forecasting baroclinic storms, and today's numerical weather prediction (NWP) of
these events has become quite accurate, even many days ahead. Importantly, baroclinic
cyclones are well resolved by today's NWP models. While baroclinic cyclones can intensify
rapidly, both the magnitude and timing of intensification are usually forecast accurately, and
uncertainties are well quantified by NWP ensembles (Leutbecher and Palmer, 2008).
By contrast, the physics of tropical cyclone intensification, involving a positive feedback between
surface winds and surface enthalpy fluxes, results in an intense, concentrated core with high
winds and heavy precipitation (which can be snow in the case of polar lows). The eyewalls of
surface flux-driven cyclones are strongly frontogenetical (Emanuel, 1997), further concentrating
wind and rain in an annulus of mesoscale dimensions. The intensity of surface flux-driven
cyclones can change very rapidly and often unpredictably, presenting a severe challenge to
forecasters. For example, Hurricane Otis of 2023 intensified from a tropical storm to a Category
5 hurricane in about 30 hours, devastating Acapulco, Mexico, with little warning from
forecasters. The small size of the core of high winds and heavy precipitation means that small
errors in the forecast position of the center of the storm can lead to large errors in local wind and
precipitation predictions. Surface flux-driven cyclones are too small to be well resolved by
today's global NWP models, and even if they were well resolved, fundamental predictability
studies show high levels of intrinsic unpredictability of rapid intensity changes (Zhang et al.,
173 2014).

The time and space scales of surface flux-driven cyclones are such that they couple strongly
with the ocean, producing near-inertial currents whose shear-driven turbulence mixes to the
surface generally (but not always) colder water from below the surface mixed layer. This has an
important (usually negative) feedback on the intensification of such cyclones. Accurate
numerical forecasting of surface flux-driven cyclones therefore requires an interactive ocean,
generally missing from today's NWP models because it is not very important for baroclinic
cyclones and because of the additional computational burden.
For these reasons, it matters (or should matter) to forecasters whether a particular development
is primarily driven by surface fluxes or by ambient baroclinity. The structural differences
described above are often detectable in satellite imagery; at the same time, such imagery is
sometimes misleading about the underlying physics. For example, classical baroclinic
development sometimes develops cloud-free eyes surrounded by convection through the warm
seclusion process, even over land, and yet may not have the intense annular concentration of
wind and rain characteristic of surface flux-driven cyclones (Tous and Romero, 2013).

## 3.    Case Studies

Armed with the conceptual CYCLOP model developed in Section 1 and modified potential
intensity given by (1), we now turn to case studies of the development of medicanes, polar lows,
a subtropical cyclone, and a Kona storm, showing that the dynamic and thermodynamic
pathways are similar. Specifically, each case developed after the formation of a deep, cold-core
cut-off cyclone in the upper troposphere that often resulted from a Rossby wave breaking event.
The lifting of the tropospheric air in response to the developing potential vorticity anomaly near
the tropopause created a deep, cold, and presumably humid column. The deep cold air over
bodies of relatively warmer water substantially elevates potential intensity, while its high relative
humidity discourages evaporatively driven convective downdrafts, which tamp down the needed
increase in boundary layer enthalpy. Low vertical wind shear near the core of the cutoff cyclone,
coupled with high potential intensity and humidity, provide an ideal embryo for tropical cyclone-
like development. The requirement for low vertical wind shear may prevent open troughs from
producing CYCLOPs, even if the potential intensity is elevated.
In what follows we focus on the cut-off cyclone evolution and the development of modified
potential intensity. In a particular case, we compare reanalysis column water vapor to that
estimated from satellite measurements. The differences between these are significant enough
to cast some doubt on the quality of reanalyzed water vapor associated with the small-scale
CYCLOP developments, thus we do not focus on water vapor even though it is known to be
important for intensification of tropical cyclones.
We also examined, but do not show here, several cases of Australian East Coast Lows
(Holland et al., 1987). Owing to the East Australian Current, the climatological potential intensity
is substantial off the southeast coast of Australia, and the CYCLOP developments we analyzed
behaved more like classical tropical transitions, with little or no role of the synoptic scale
dynamics in enhancing the existing potential intensity. We suspect that there may be other
cases in which the latter process was important, but did not conduct a search for such cases. It
is clear that in many of these cases surface heat fluxes were important in driving the cyclone
(Cavicchia et al., 2019). We also examined a small number of subtropical cyclones that
developed in the South Atlantic, but as with the Australian cases, they resembled classical
tropical transitions, though again we suspect there may be CYCLOP cases there as well.
As with tropical cyclones, there are variations on the theme of CYCLOPs, and we explore these
in the closing sections.
3.1 Medicane Celeno of January, 1995
On average, 1.5 medicanes are observed annually in the Mediterranean (Cavicchia et al., 2014;
Nastos et al., 2018; Romero and Emanuel, 2013; Zhang et al., 2021) and, more rarely, similar
storms are observed over the Black Sea (Yarovaya et al., 2008). We begin with a system,
Medicane Celeno, that reached maturity on 16 January, 1995, shown in infrared satellite
imagery in Figure 2.

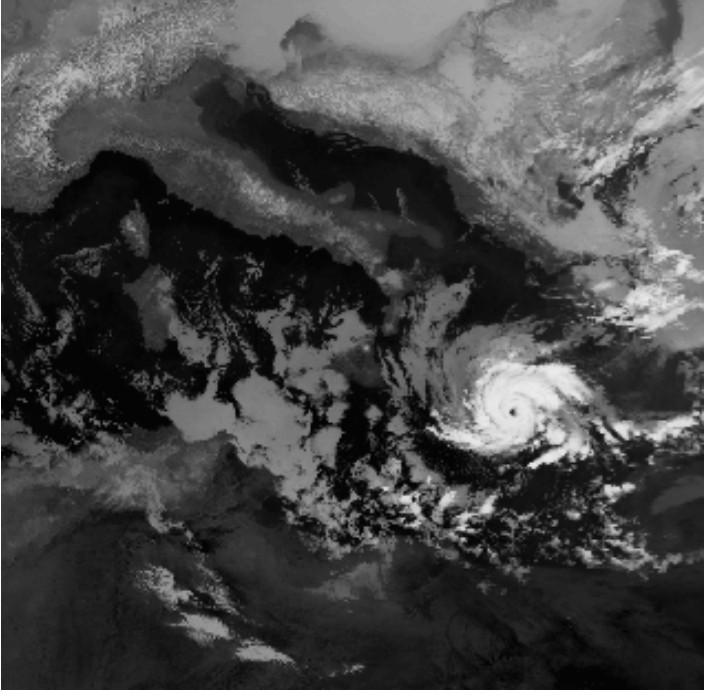


*Figure 2: Infrared satellite image of Medicane Celeno in the central Mediterranean, at 09:06 UTC 16 January 1995.*
*(Image credit: NOAA, 1995)*

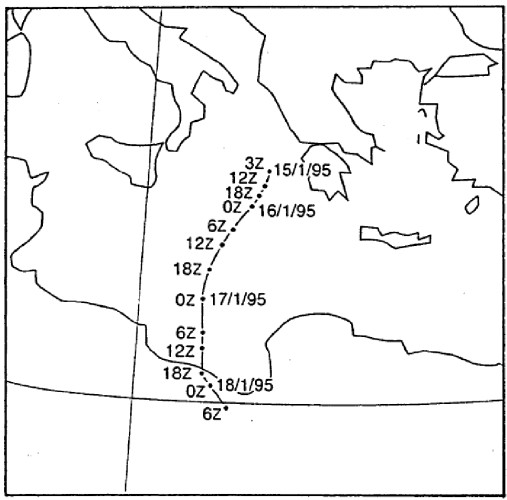


*Figure 3: Track of the surface center of the cyclone between 03:00 UTC on January 15 and 06:00 on January 18, 1995, derived from satellite images (Pytharoulis et al.,1999)*


Figure 3 shows the track of the surface center of the cyclone, which developed between Greece
and Sicily and made landfall in Libya. A detailed description of this medicane is provided by
Pytharoulis et al. (1999).
The evolution of the system in increments of 24 hours, beginning on 00 UTC, 12 January and
ending at 00 UTC 15 January, is shown in Figure 4. This sequence, showing the 400 hPa and
950 hPa geopotential heights, covers the period leading up to the CYCLOP development. The
evolution of the upper tropospheric (400 hPa) height field shows a classic Rossby wave
breaking event (McIntyre and Palmer, 1983) in which an eastward-moving baroclinic Rossby
wave at higher latitudes amplifies and irreversibly breaks to the south and west, finally forming a
cut-off cyclone. It is important to note that the formation of a cut-off cyclone in the upper
troposphere involves a local conversion from kinetic to potential energy, with deep cooling
associated with dynamically forced ascent (Gan and Piva, 2016). To the southeast of this
trough, a weak, broad area of low geopotential height develops at 950 hPa mostly over land in a
region of low-level warm advection. As the cold air under the cut-off low is gradually heated by
the underlying sea, a broad surface cyclone develops by 00 UTC on January 14th. At around
this time, the feedback between surface wind and surface enthalpy fluxes is strong enough to
develop a tight inner warm core (warm, that is, relative to the surrounding cold pool, not
necessarily to the unperturbed larger scale environment) and by 00 UTC on January 15th, it is
noticeable just to the west of northwestern Greece (Figure 5d). At the same time, the upper cold
core weakens, perhaps aided by the strong heating from the surface transferred aloft by deep
convection.

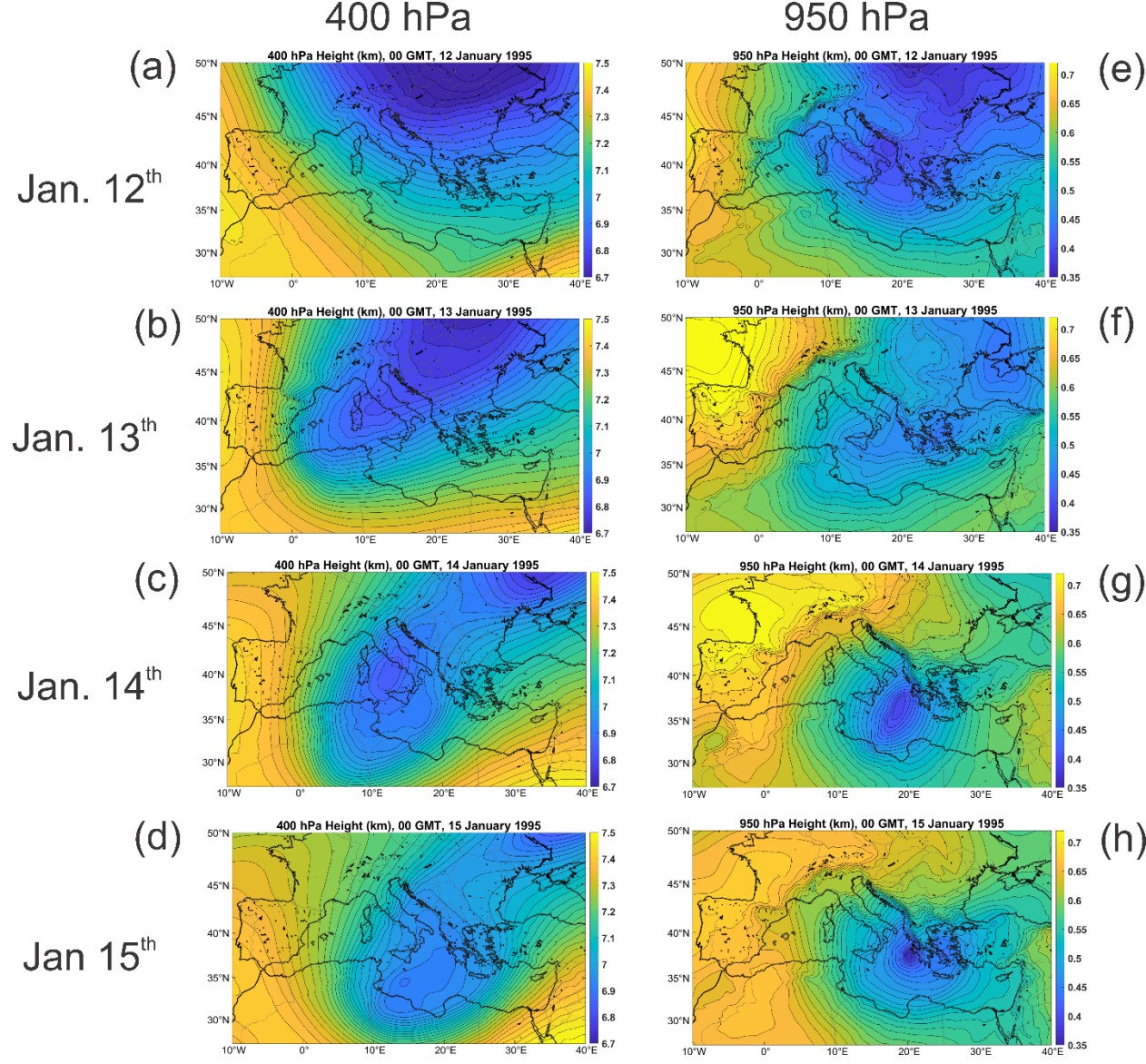

*Figure 4: Evolution of the 400 hPa (left) and 950 hPa (right) geopotential heights in 24-hour increments, from 00 UTC 12 January to 00 UTC 15 January, 1995. These fields are from ERA5 reanalysis. The 400 hPa heights span from 6.7 to 7.5 km at increments of 26.67 m, while the 950 hPa heights range from 350 m to 730 m at increments of 12.67 m.*

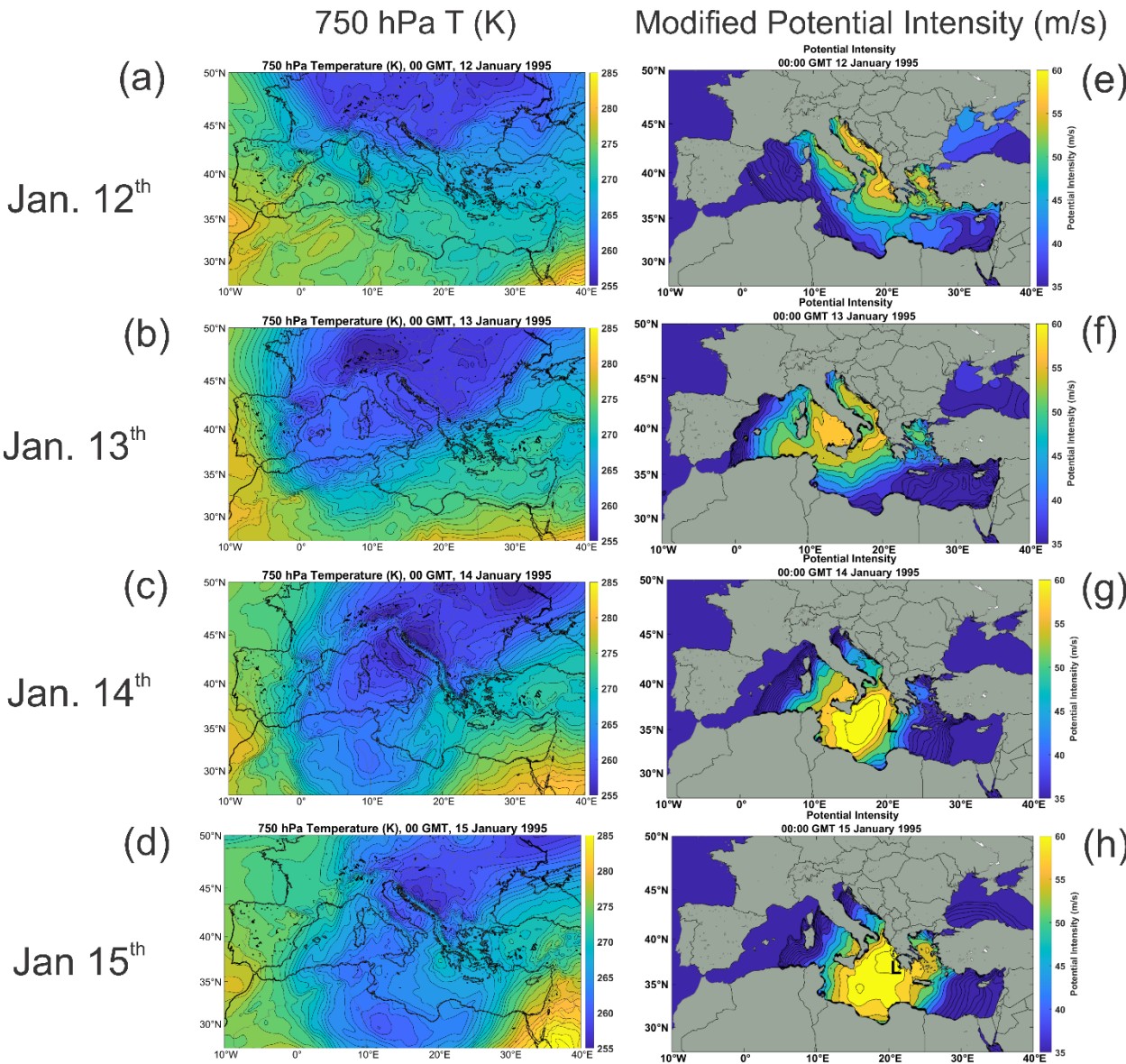

261

*Figure 5: 750 hPa temperature (K; left) and modified potential intensity given by equation (1) (m/s, right) at 00 UTC on January 12<sup>th</sup> – 15<sup>th</sup> (top to bottom) 1995.  The 750 hPa temperature spans from 255 K to 285 K at increments of 1 K, while the modified potential intensity ranges from 35 to 60 ms<sup>-1</sup> at increments of 4 ms<sup>-1</sup>. The black "L"s in panels g) and h) show the positions of the surface cyclone. From ERA5 reanalysis.*

Figure 5 shows the corresponding evolution of the 750 hPa temperature and the modified potential intensity, $V_{pm}$, given by (1).  The latter is only shown in the range of $35\,ms^{-1}$ to $60\,ms^{-1}$. Experience shows that tropical cyclone genesis is rare when the potential intensity is less than about $35\,ms^{-1}$ (Emanuel, 2010).

On January 12th, relatively small values of $V_{pm}$ are evident in the eastern and northern
Mediterranean, but with higher values developing over the northern Adriatic as cold air aloft
creeps in from the north. As the upper tropospheric Rossby wave breaks and a cut-off cyclone
develops over the central Mediterranean, $V_{pm}$ increases greatly during the 13th and 14th, with
peak values greater than $60\ ms^{-1}$. The CYCLOP develops in a region of high $V_{pm}$, though not
where the highest values occur, until the 15th at which time the mature CYCLOP location
corresponds to that of the highest $V_{pm}$. Note that a lower warm core is not obvious at 750 hPa
until January 15th and that it occurs on a somewhat smaller scale than the cut-off cyclone.
Figure A1 of the Appendix shows the individual contributions, on January 14th and 15th, of the
usual potential intensity (first term on the right side of (1)) and the correction for column warming
(second term on right of (1)) to the full potential intensity. (The contribution of column warming
was negligible before the 14th.) Note that the full potential intensity shown in Figure 5 *g* and *h* is
not the arithmetic sum of the two contributions but rather their Euclidean norm. The contribution
of the column warming, as measured by the surface pressure anomaly, is quite localized around
the CYCLOP and is located eastward of the maximum of the usual potential intensity.
As the CYCLOP moves southward, it maintains a tight inner warm core until after landfall (not
shown here) and the modified potential intensity, $V_{pm}$, over the south-central Mediterranean
diminishes rapidly, with peak values of only about $50\ ms^{-1}$ by 00 UTC on January 16th. One
potentially important difference between CYCLOP development and tropical transition is that in
the former case, the volume of air with appreciable potential intensity is limited. It is possible
that, by warming the whole tropospheric column relative to the surface, the enhanced surface
fluxes associated with the CYCLOP, acting through deep convection, substantially diminish the
magnitude and/or volume of the high $V_{pm}$ air, serving to limit the lifetime of the CYCLOP. This
was the case in the axisymmetric numerical simulations of Emanuel (2005).
It is clear that this CYCLOP development occurred on time and space scales appreciably
smaller than those of the rather weak, synoptic-scale cyclogenesis resulting from the interaction
of the positive PV perturbation near the tropopause with a low-level temperature gradient. It is
also clear that the strong cooling of the troposphere in response to the development and
southward migration of the cutoff cyclone aloft was instrumental in bringing about the high
potential intensity necessary to activate the WISHE process. Therefore, the Rossby wave
breaking did much more than trigger cyclogenesis; it provided the necessary potential intensity
for the CYCLOP development. Here we draw a distinction from tropical transition, in which, in
most cases, the pre-existing potential intensity suffices to maintain a surface flux-driven cyclone.
We emphasize again that there is a gray area here consisting of flux-driven cyclones in which
the existing reservoir of potential intensity is enhanced to some degree by the local synoptic-
scale dynamics.

## 3.2  Medicane of December 2005
A medicane, unofficially known as Zeo, formed on December 14th 2005 off the coast of Tunisia
and then moved eastward across a large stretch of the Mediterranean. Figure 6 shows a
satellite image of this medicane on December 15th.  This cyclone has been the subject of
several intensive studies (e.g. Fita and Flaounas, 2018; Miglietta and Rotunno, 2019).
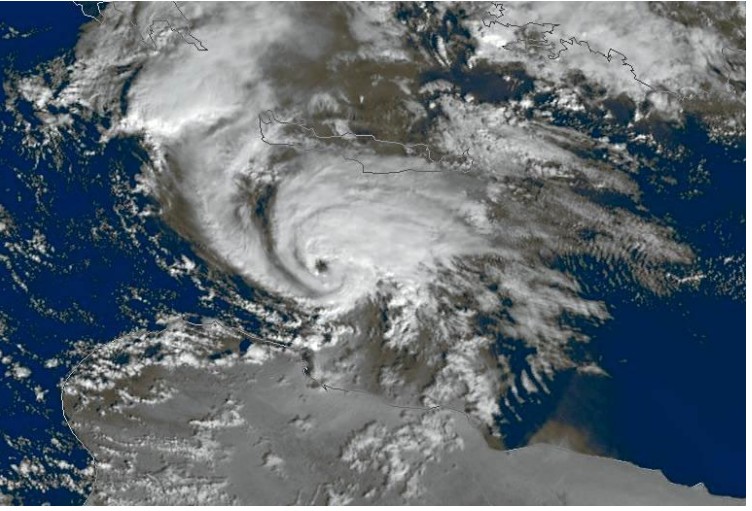
*Figure 6:  Medicane Zeo between Crete and Libya on 15 December 2005. Image from NASA.*
Like Medicane Celeno, Medicane Zeo formed as a result of a Rossby wave breaking event, as
illustrated in Figure 7. On December 8th (panel a), a deep trough is digging southward over
eastern Europe, but by the 9th (panel b) it has split in two, with the westward half breaking
southwestward over Germany and Switzerland. By the 11th (panel d), this local minimum is
located over Tunisia and on the 12th (panel e) it is more or less completely cut off from the main
westerly jet. It subsequently oscillates over Tunisia and Algeria, before moving slowly eastward
over the Mediterranean, as shown in Figure 8.
In response to the cutoff cyclone development, a broad, synoptic-scale surface cyclone formed
to the east of the upper cyclone over the deserts of Libya during December 12th and 13th (not
shown here). The poorly defined center of this system drifted northward, on collision course with
the upper cyclone, which was drifting eastward over Tunisia by late on the 13th. As the surface
center moved out over the Mediterranean early on the 13th, rapid development ensued and a
mature CYCLOP was evident by 00 UTC on the 14th (Figure 8c). The system began to move
eastward and reached peak intensity on the 14th (left panels of Figure 8). The CYCLOP moved
eastward, along with its parent upper cyclone, and dissipated in the far eastern Mediterranean
on the 16th (not shown).


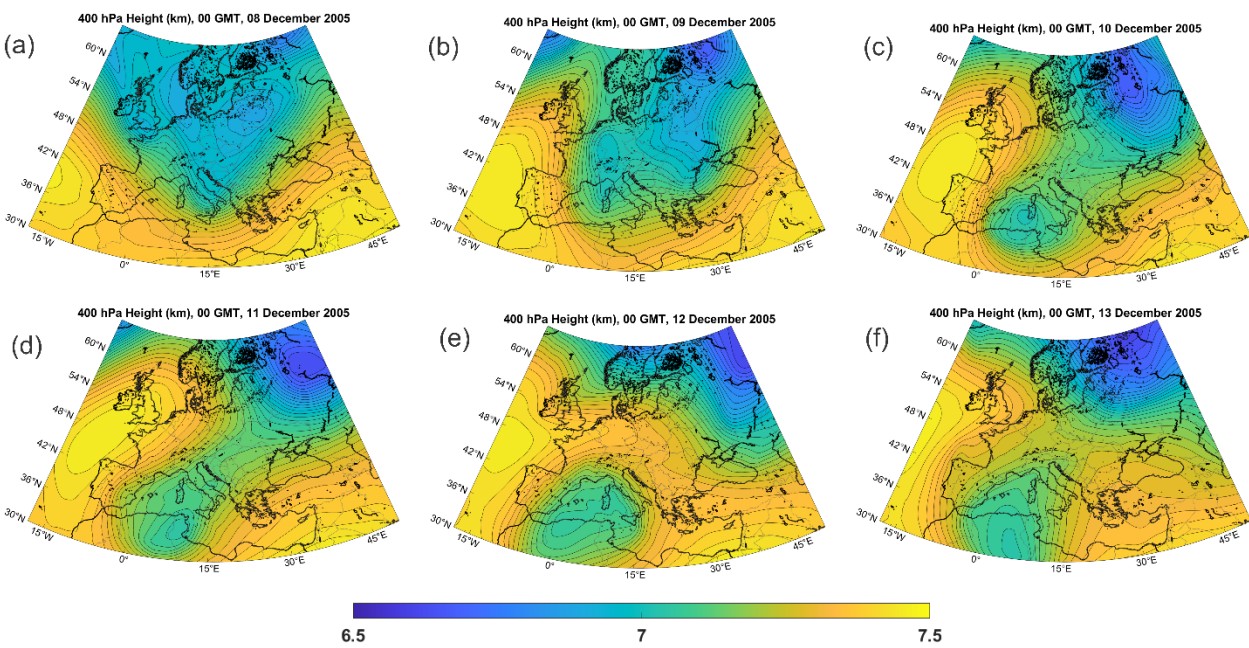


Figure 7: 400 hPa geopotential heights, ranging from 6.5 km to 7.5 km (color scale identical for all panels) at 00 UTC on December 8th (a), 9th (b), 10th (c), 11th (d), 12th (e), and 13th (f), 2005. From ERA-5 reanalysis. Contour interval is 33.3 m.

The evolutions of 750 hPa temperature and $V_{pm}$ on the 14th and 15th of December are shown in
Figure 9. In this case, there were large meridional gradients of sea surface temperature across
the Mediterranean, ranging from over 20°C in the far south to less than 12°C in the northern
reaches of the Adriatic and western Mediterranean (not shown here). As the cold pool moved
southwestward and deepened, only low values of $V_{pm}$ are present over the Mediterranean, but
beginning on December 10th, higher values developed over the Gulf of Sidra, east of Tunisia,
and by the 14th (Figure 9c) had reached at least 80 ms$^{-1}$, enabling the formation of the CYCLOP.
As in the Celeno case, $V_{pm}$ values diminish thereafter, possibly because of the warming of the
column by the surface enthalpy flux.

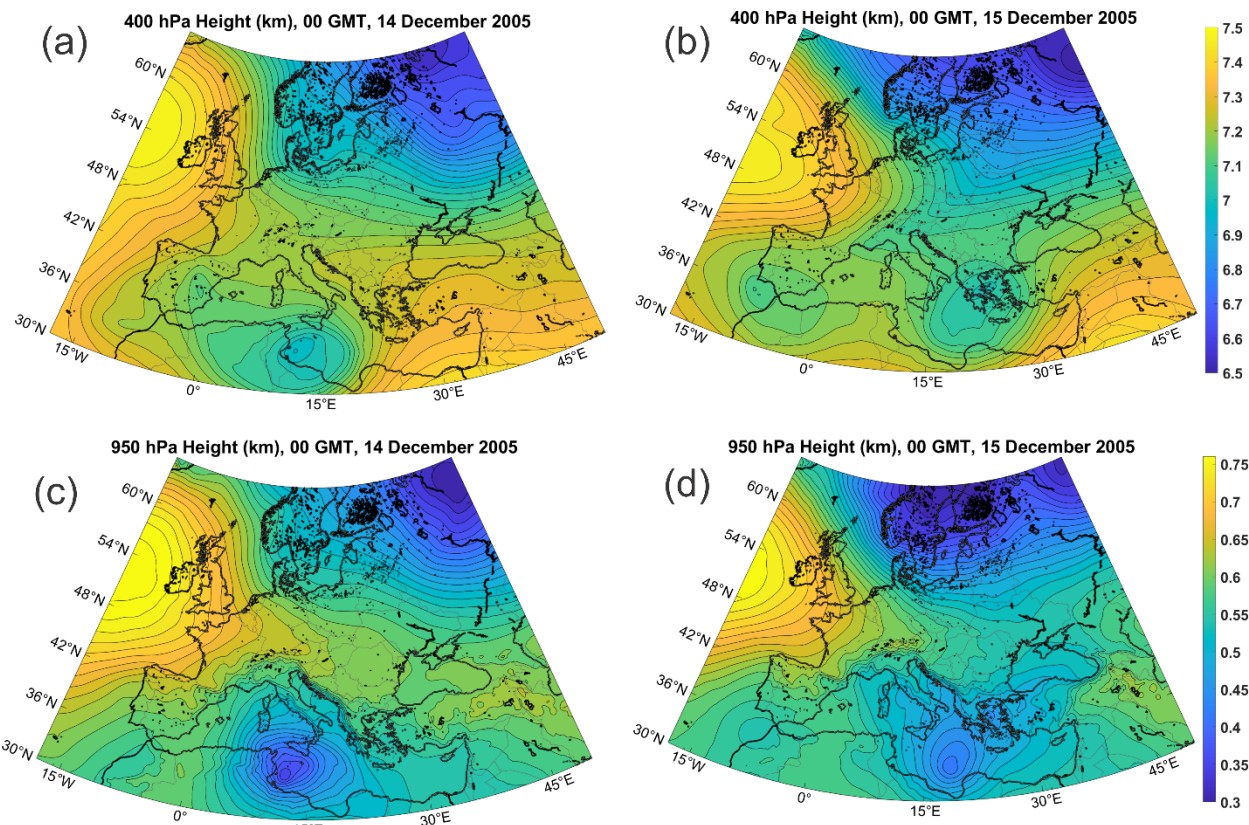


*Figure 8:  Evolution of 400 hPa geopotential height, ranging from 6.5 to 7.5 km at contour interval of 36.7 m (a and b),*
*and 950 hPa geopotential height, ranging from 250 to 750 m at contour interval of 16.67 m (c and d) at 00 UTC in*
*December 14th (left) and 15th (right), 2005. The color scales of (a) and (b) are identical, as are the color scales of (c)*
*and (d). From ERA-5 reanalysis.*


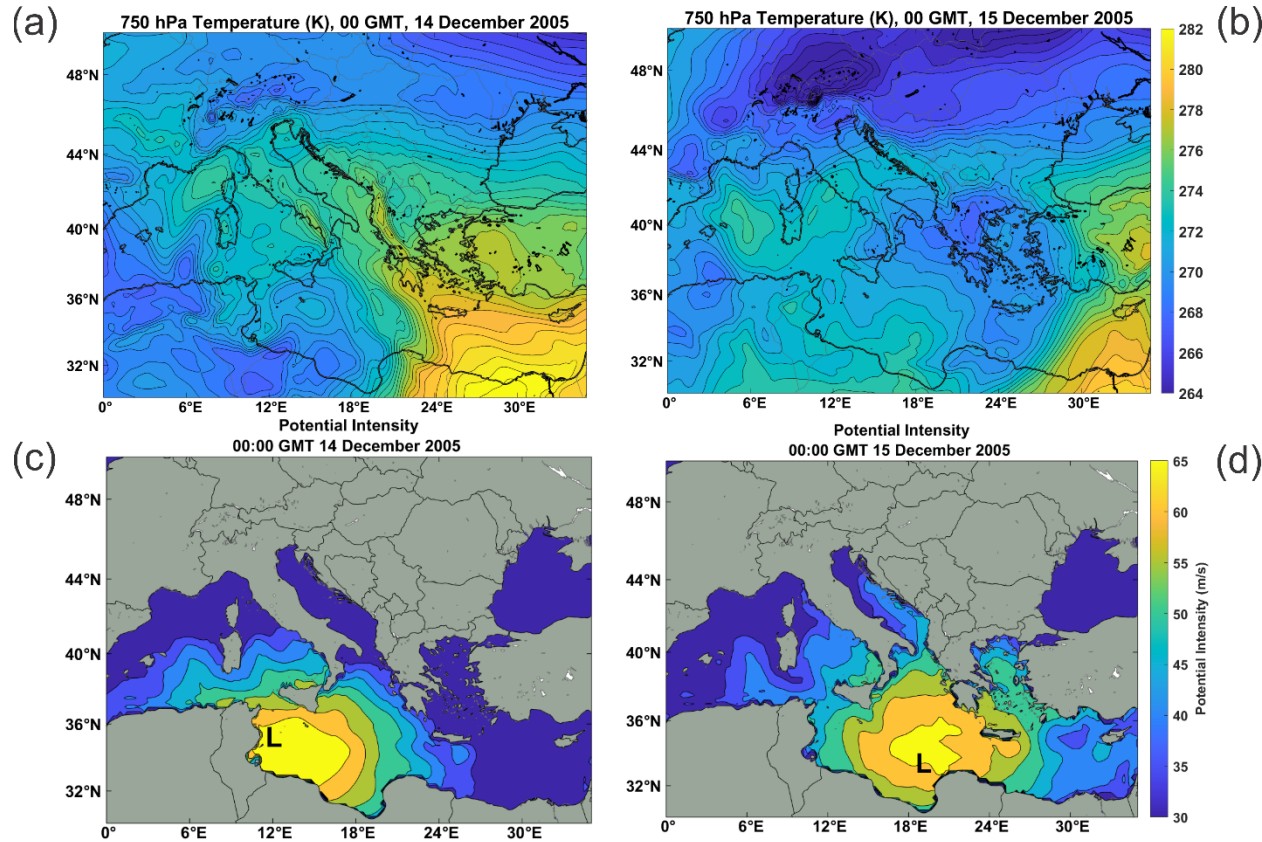

Figure 9: 750 hPa temperature at increments of 0.6 K (a-b) and modified potential intensity from 35 m s$^{-1}$ to 65 m s$^{-1}$ at increments of 5 m s$^{-1}$ (c-d) at 00 UTC on December 14$^{th}$ (a,c) and December 15$^{th}$ (b,d) 2005. From ERA-5 reanalyses. The "L"s in c) and d) indicate the position of the 950 hPa cyclone center.

The 750 hPa temperature field (Figure 9a) shows a complex pattern of positive temperature perturbation near the position of the surface low, not nearly as focused as in the case of Medicane Celeno. As suggested by Fita and Flaounas (2018), part of the positive temperature anomaly associated with the cyclone may have resulted from a warm seclusion. Yet 24 hours later (Figure 9b) the warm anomaly near the cyclone center over the eastern Gulf of Sidra appears to have been advected from the south rather than the north. It is possible that the reanalysis did not capture the full physics of this particular medicane.

Figure 10 compares total column water retrieved from satellite microwave and near-infrared imagers (a) to that from the ERA5 reanalysis (b). While there is some broad agreement between the two estimates, the ERA5 underestimates column water in the critical region just east of Tunisia and overestimates it in an arc extending from eastern Sicily southeastward to the Libyan coast. Cyclop intensity, in analogy to tropical cyclone intensity, should be highly sensitive to low- and mid-level moisture in the mesoscale inner core region, which may be under-resolved and otherwise not well simulated by global NWP models. For this reason, we do not routinely show reanalysis of water vapor in this paper. The limitations of analyzed and forecast water vapor

may constitute an important constraint on CYCLOP predictability and is a worthy subject for
future research.

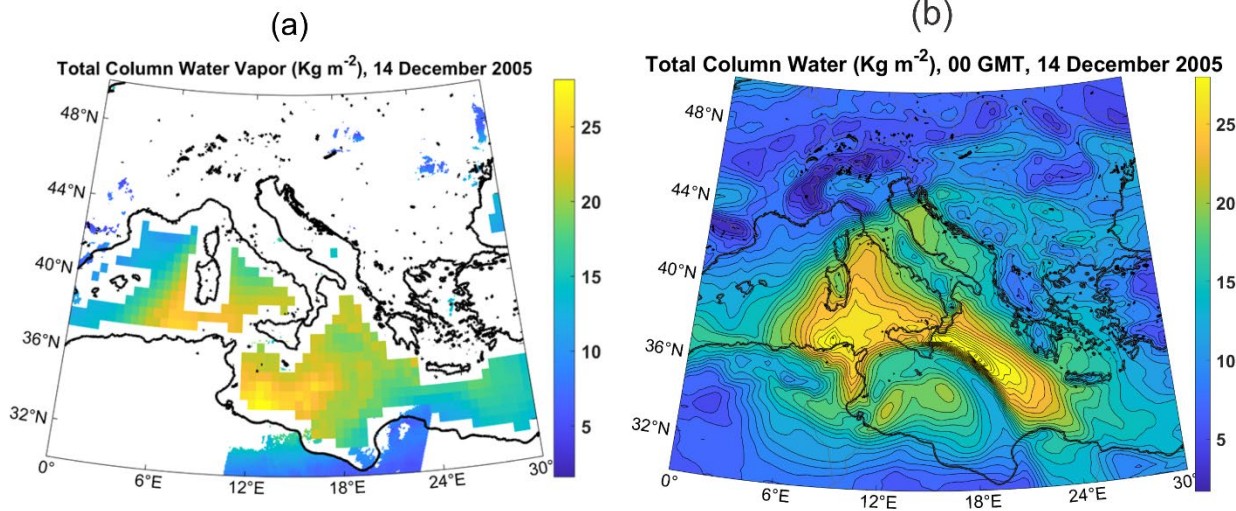

*Figure 10:  Total column water vapor (Kg m$^{-2}$) at 00 UTC on 14 December 2005, derived from microwave and near*
*infrared imagers (a) and ERA5 reanalysis (b). In (b) the contour increment is 0.9 Kg m$^{-2}$.*

## 3.3  Polar low of February, 1987

Closed upper tropospheric lows also provide favorable environments for CYCLOP development
at very high latitudes in locations where there is open water. These usually form poleward of the
mid-latitude jet, where quasi-balanced dynamics can be quite different from those operating at
lower latitudes. Figure 11 is an infrared image of a polar low that formed just south of Svalbard
on February 25[th], 1987, and tracked southward, making landfall on the north coast of Norway on
the 27[th]. This system was studied extensively by Nordeng and Rasmussen (1992).

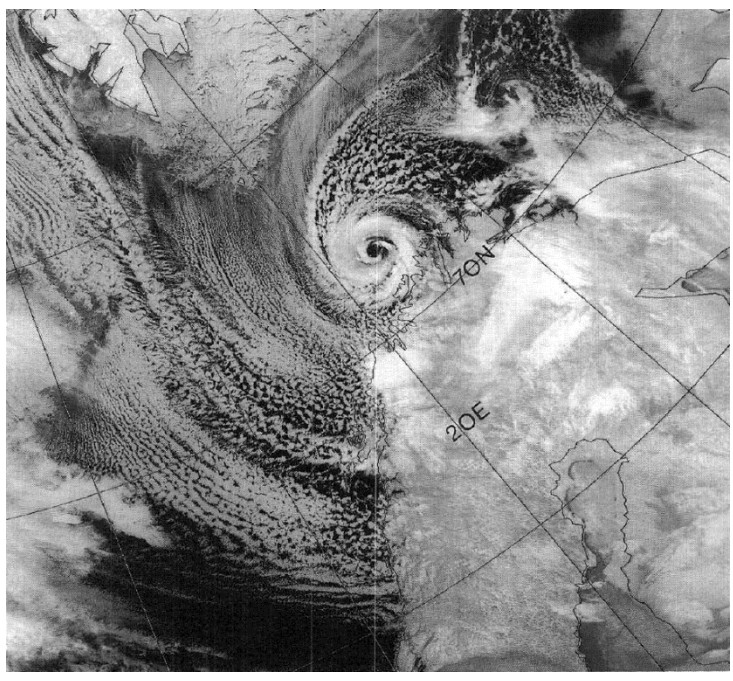


*Figure 11: NOAA 9 satellite infrared image (channel 4) of a polar low just north of Norway at 08:31 UTC on 27*
*February 1987.*
As with some medicanes, polar lows develop in strongly convecting air masses when cold air
moves out over relatively warm water. The adjective "relatively" is crucial here; with polar lows
the sea surface temperature is often only marginally above the freezing point of saltwater.
Figure 12 shows the distribution of sea surface temperature on February 25[th], with the uniform
dark blue areas denoting regions of sea ice cover. The polar low shown in Figure 11 develops
when deep cold air moves southward over open water, as shown in Figure 13.
In this case, it is not clear whether one can describe what happens in the upper troposphere
(top row of Figure 13) as a Rossby wave breaking event. Instead, what we see is a complex
rearrangement of the tropospheric winter polar vortex, as a ridge building over North America
breaks and forms an anticyclone over the North Pole. This complex rearrangement results in the
formation of a deep cutoff low just south of Svalbard by the 26[th], which then moves southward
over Norway by the 27[th]. The polar low is barely visible in the 950 hPa height field on the 25[th]
(middle row of Figure 13), but intensifies rapidly as it moves over progressively warmer water,
reaching maturity before landfall on the 27[th].

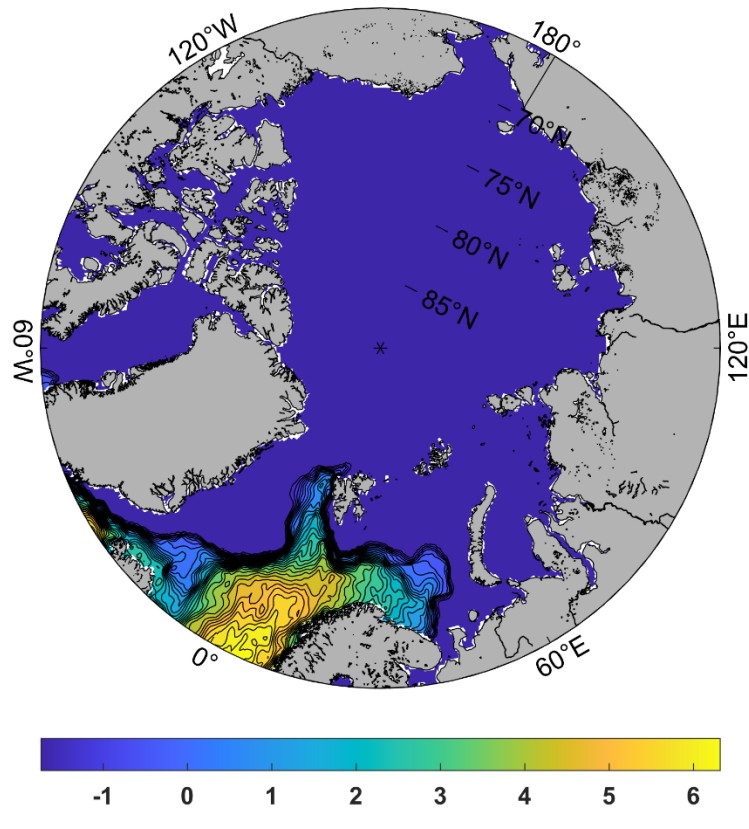


Figure 12:  Sea surface temperature (ºC) in intervals of 0.3ºC at 02:00 UTC on 25 February, 1987, from ERA5
reanalysis. Dark blue areas denote regions of sea ice. From ERA-5 reanalyses.

The evolution of the $V_{pm}$ field is shown in the bottom row of Figure 13. The potential intensity
increases rapidly south of Svalbard as the cut-off cyclone moves out over open water.

Under these conditions, most of the surface flux that drives the CYCLOP is in the form of
sensible, rather than latent heat flux, and the background state has a nearly dry (rather than
moist) adiabatic lapse rate. As shown by Cronin and Chavas (2019) and Velez-Pardo and
Cronin (2023), surface flux-driven cyclones can develop in perfectly dry convecting
environments, though they generally reach smaller fractions of their potential intensity and lack
the long tail of the radial profile of azimuthal winds that is a consequence of the dry stratification
resulting from background moist convection (Chavas and Emanuel, 2014). They also have
larger eyes relative to their overall diameters. Given the low temperatures at which they occur,
polar lows may be as close to the Cronin-Chavas dry limit as one might expect to see in Earth's
climate.

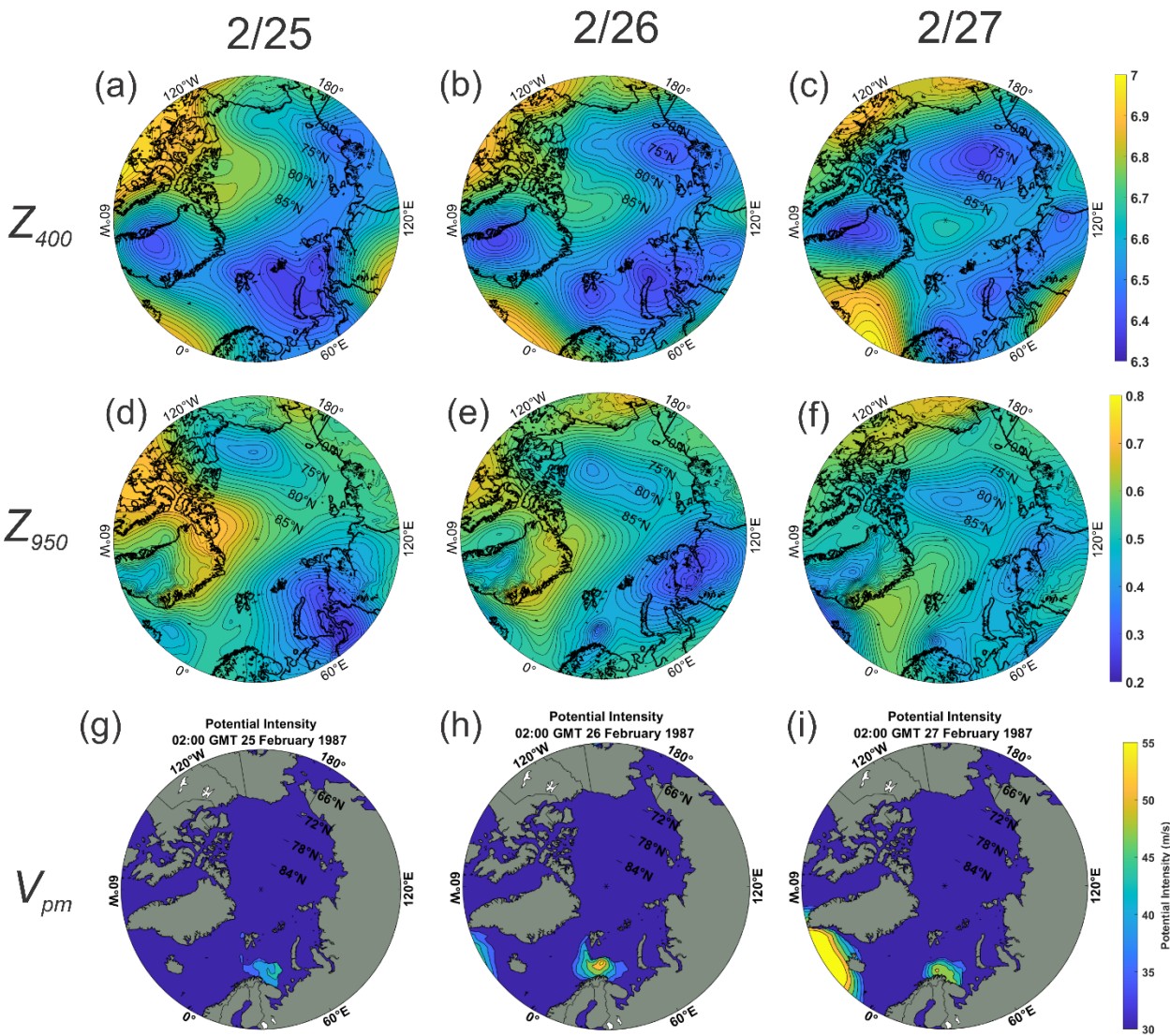

*Figure 13: 400 hPa geopotential height (km)t at intervals of 23.3 m (a-c), 950 hPa geopotential height (km) in intervals of 20 m (d-f), and $V_{pm}$ (ms⁻¹) in intervals of 4 ms⁻¹ (g-i), at 02:00 UTC on February 25ᵗʰ (left), 26ᵗʰ (center), and 27ᵗʰ (right), 1987. From ERA-5 reanalysis.*

One interesting feature of polar waters in winter is that the thermal stratification is sometimes reversed from normal, with warmer waters lying beneath cold surface waters. This is made possible, in part, by strong salinity stratification, that keeps the cold water from mixing with the warmer waters below. Therefore, it is possible for polar lows to generate warm, rather than cold, wakes, and this would feed back positively on their intensity. This seems to happen in roughly half the documented cases of polar lows in the Nordic seas (Tomita and Tanaka, 2024).

## 3.4   Polar low over the Sea of Japan, December, 2009
Polar lows are not uncommon in the Sea of Japan, forming when deep, cold air masses from
Eurasia flow out over the relatively warm ocean. They are frequent enough to warrant a
climatology (Yanase and co-authors, 2016). A satellite image of one such storm is shown in
Figure 14.

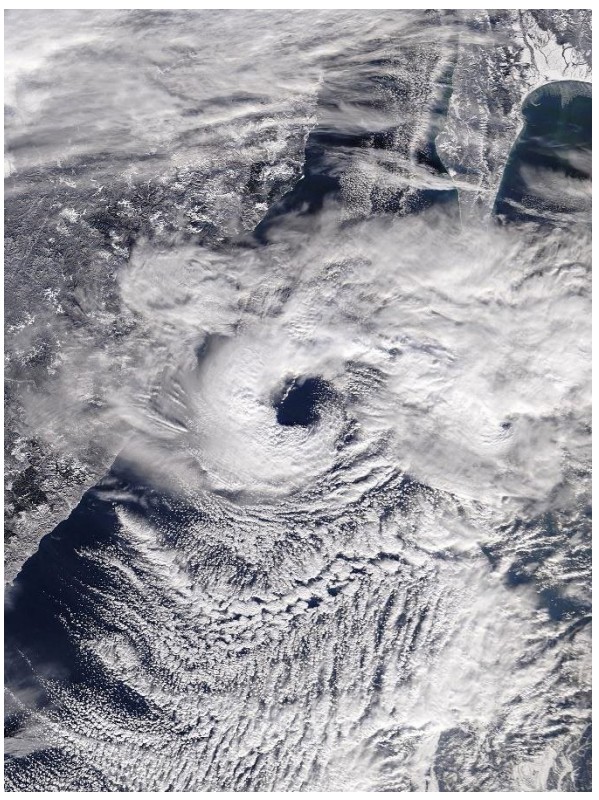


*Figure 14: Polar low over the northern Sea of Japan, 02:13 UTC 20 December, 2009 as captured by the MODIS*
*imager on NASA's Terra satellite.*
The cyclone traveled almost due south from this point, striking the Hokkaido region of Japan,
near Sapporo, with gale-force winds and heavy snow. As with other CYCLOPs, it formed in an
environment of deep convection under a cold low aloft.
The development of the cutoff cyclone aloft was complex, as shown by the sequence of 400
hPa maps displayed in Figure 15. These are 00 UTC charts at 1-day intervals beginning on
December 11[th] and ending on the 20[th], about the time of the image in Figure 14. A large polar
vortex is centered in northern central Russia on the 11[th] but sheds a child low southeastward on
the 12[th] and 13[th], becoming almost completely cutoff on the 14[th]. The parent low drifts westward
during this time. The newly formed cutoff cyclone meanders around in isolation from the 15[th]
through the 17[th], but becomes wrapped up with a system propagating into the domain from the
east on the 18[th]. By the 20[th], a small-scale cutoff cyclone is drifting southward over the northern
Sea of Japan, and it is this upper cutoff that spawns the polar low.

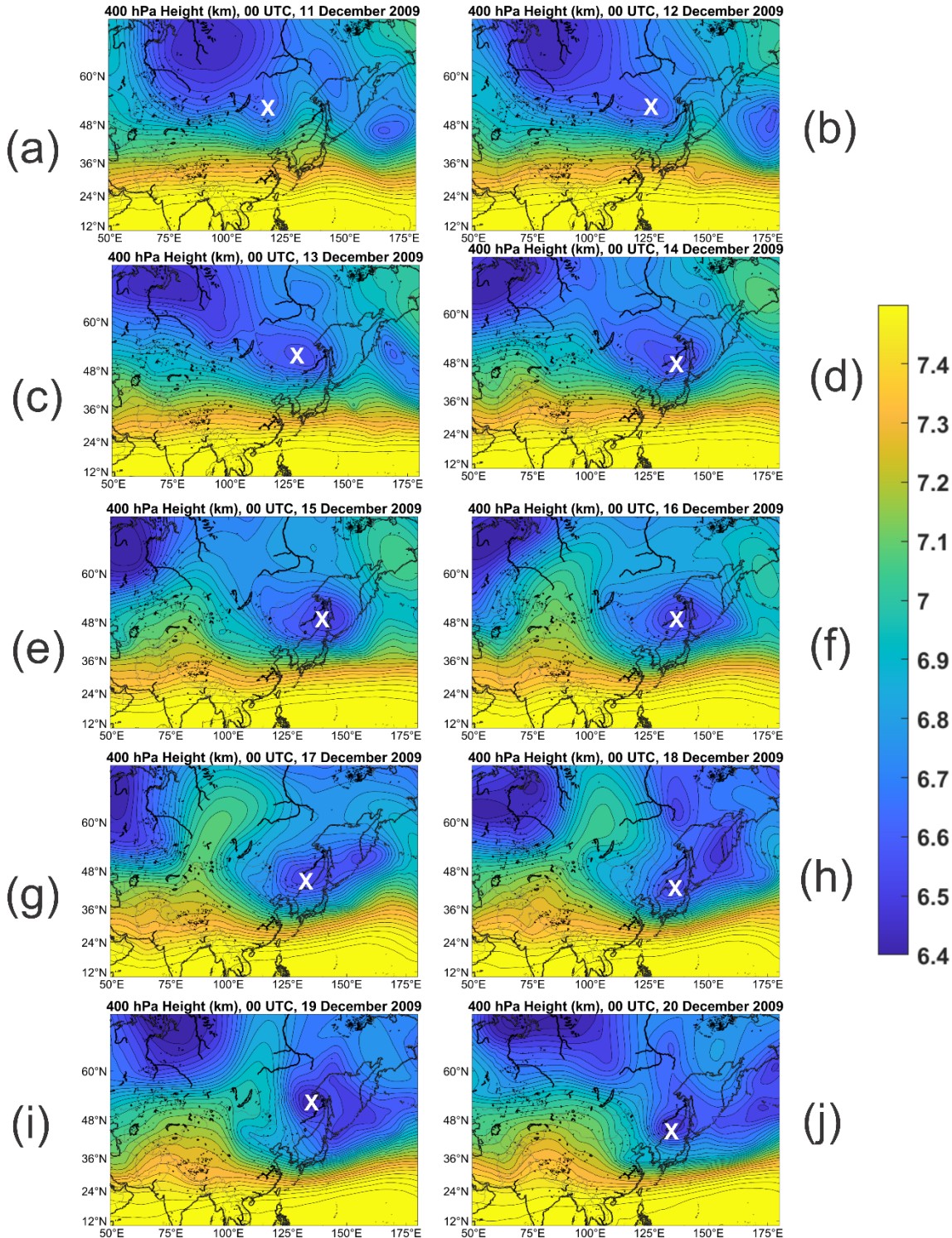


Figure 15: Sequence of 400 hPa geopotential height (km) charts at 1-day intervals from 00 UTC on December 11[th]
(a) to 00 UTC on December 20[th] (j), 2009. The charts span from 0º to 180º longitude and from 10º to 70º latitude, and
the contour increment is 36.7 m. The white crosses mark the center of the 400 hPa cyclone that was ultimately
associated with the polar low.  From ERA-5 reanalysis.

The 950 hPa height and the $V_{pm}$ fields at 00 UTC on December 20th are shown in Figure 16. At
this time, the surface low is developing rapidly and moving southward into a region of high
potential intensity. The latter reaches a maximum near the northwest coast of Japan, where the
sea surface temperatures are larger. As with the two medicane cases and the other polar low
case, the CYCLOP develops in a place where the potential intensity values are normally too low
for surface flux-driven cyclones but for which the required potential intensity is created by the
movement of a deep cold cyclone aloft over relatively warm water. The reanalysis 750 hPa
temperature (not shown) presents a local maximum at the location of the surface cyclone at this
time.

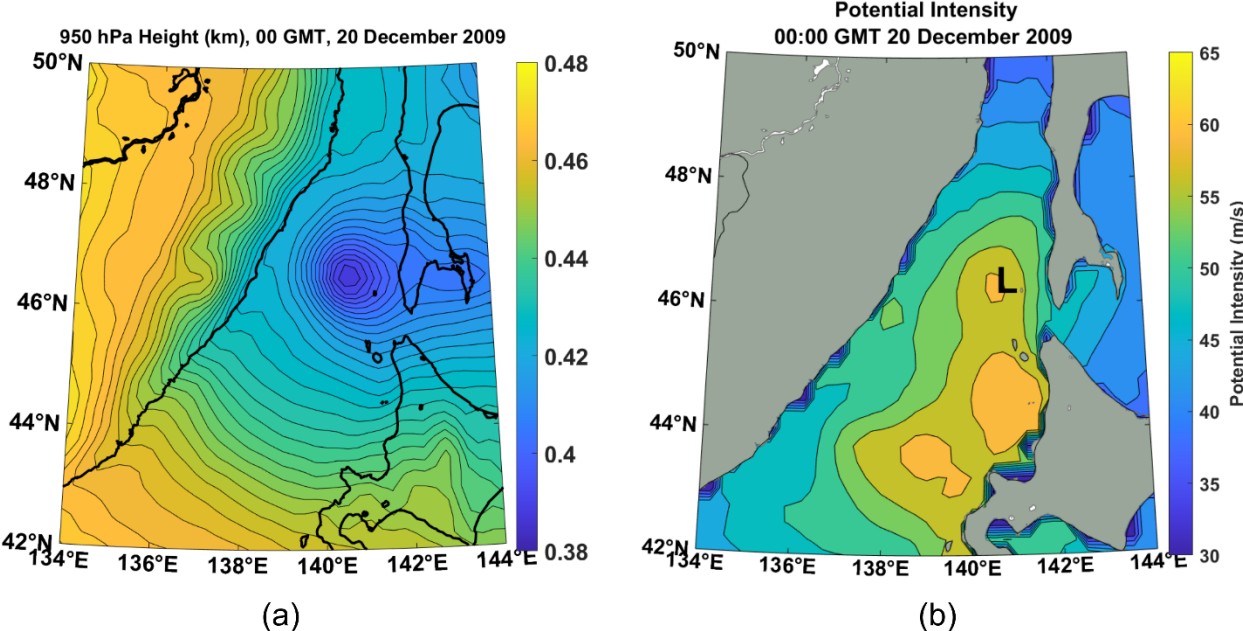

464                             (a)                                             (b)

*Figure 16: 950 hPa geopotential height (a; km) at intervals of 3 m, and $V_{pm}$ at intervals of 3 m s$^{-1}$ (b), at 00 UTC on*
*20 December, 2009. In (b), the "L" marks the satellite-derived surface cyclone center. From ERA-5 reanalysis.*

3.5  The subtropical cyclone of January, 2023
The term "subtropical cyclone" has been used to describe a variety of surface flux-assisted
cyclonic storms that do not strictly meet the definition of a tropical cyclone. The term has an
official definition in the North Atlantic[5] but is used occasionally elsewhere, especially in regions
where terms like medicane, polar low, and Kona storm do not apply, such as the South Atlantic
(Evans and Braun, 2012; Gozzo et al., 2014). Here we will use the term to designate CYCLOPs
in the sub-arctic North Atlantic; that is, surface flux-powered cyclones that develop in regions

---

[5] The National Hurricane Center defines a "subtropical cyclone" as "a non-frontal low-pressure system
that has characteristics of both tropical and extratropical cyclones", but in the past has also used the
terms "hybrid storm" and "neutercane".

and times whose climatological thermodynamic potential is small or zero, that would not be
called polar lows owing to their latitude. This usage may not be consistent with other definitions.
The point here is to show that CYCLOPs can occur in the North Atlantic and we can safely refer
to these as CYCLOPs whether or not they meet some definition of "subtropical cyclone".
Figure 17 displays a visible satellite image of a subtropical cyclone over the western North
Atlantic on 16 January, 2023. It resembles the medicanes and polar lows described previously,
and like them, formed under a cutoff cyclone aloft.

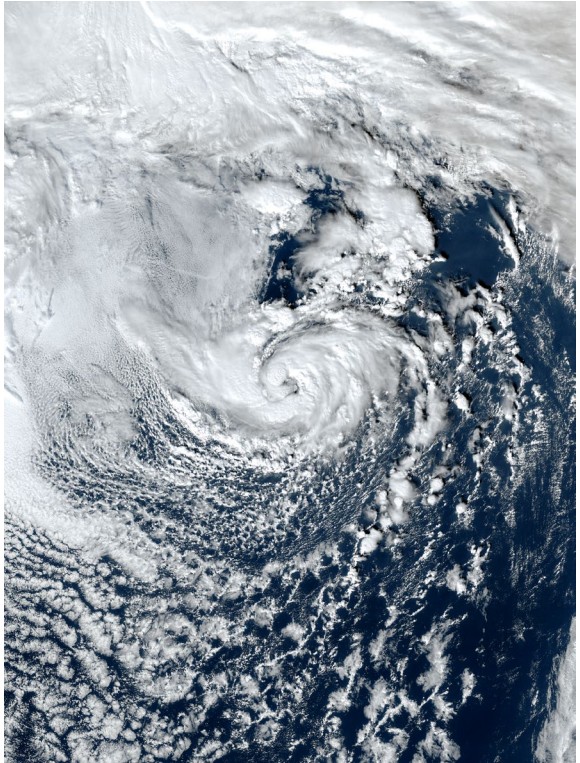


*Figure 17: Subtropical cyclone over the western North Atlantic, 18:20 UTC, 16 January 2023. NOAA geostationary*
*satellite image.*

The formation of the cutoff cyclone aloft is shown in Figure 18. A deep trough advances slowly
eastward over eastern North America and partially cuts off on the 14th. As the associated cold
pool and region of light shear migrate out over the warm waters south of the Gulf Stream, a
CYCLOP forms and intensifies with peak winds of around 60 kts at around 00 UTC on the 17th
(Cangliosi et al., 2023).  Note also the anticyclonic wave breaking event to the east of the
surface cyclone development, in the eastern half of panels 18b-d.
The evolutions of the 950 hPa and associated $V_{pm}$ fields are displayed in Figure 19. (Note the
smaller scale around the developing surface cyclone, compared to Figure 18.) On January 14th,
the only appreciably large values of $V_{pm}$ are associated with the Gulf Stream and in the far
southwestern portion of the domain. The 950 hPa height field shows a broad trough associated
with the baroclinic wave moving slowly eastward off the U.S. east coast. But as the upper cold
cyclone moves out over warmer water on the 15$^{th}$, large $V_{pm}$ develops south of the Gulf Stream,
and a closed and more intense surface cyclone develops under the lowest 400 hPa heights and
over the region with higher $V_{pm}$ values.

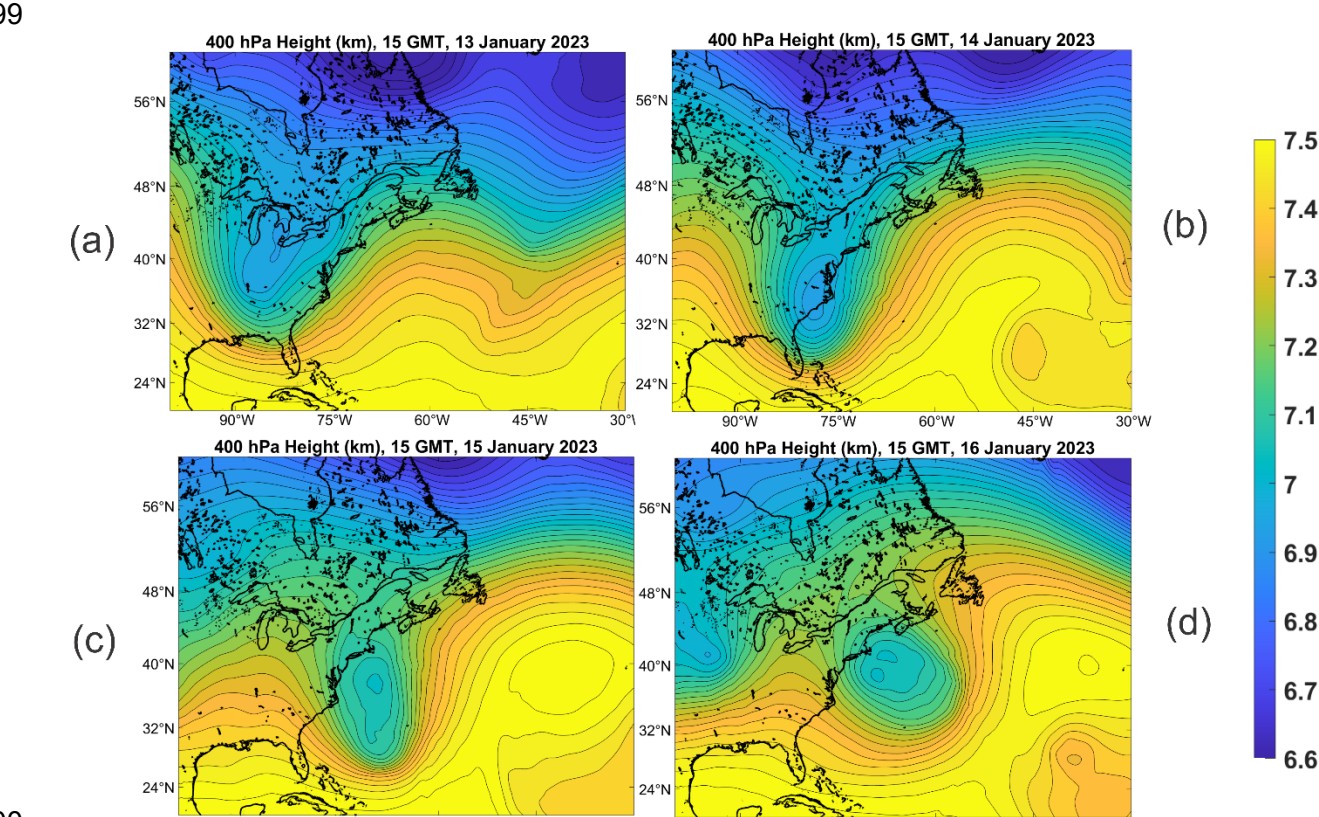


*Figure 18:  400 hPa geopotential height (km) at intervals of 30 m, at 15:00 UTC on January 13$^{th}$ (a), 14$^{th}$ (b), 15$^{th}$ (c),*
*and 16$^{th}$ (d), 2023. From ERA-5 reanalysis.*

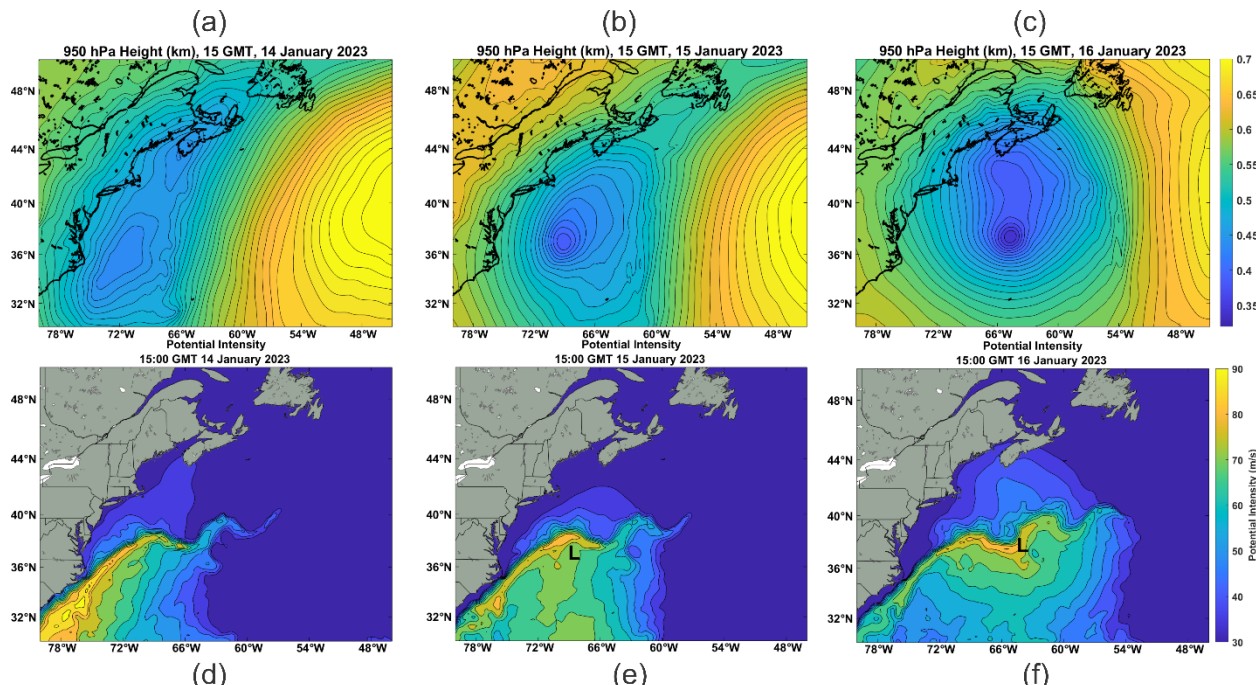

(d)           (e)           (f)


*Figure 19: The 950 hPa geopotential height (km) at 15:00 UTC on January 14th (a), 15th (b), 16th (c); the $V_{pm}$ field in*
*increments of 5 ms⁻¹, on January 14th (d), 15th (e), and 16th(f). In (e) and (f) the "L" shows the position of the 950 hPa*
*cyclone center at the time of the chart. From ERA-5 reanalysis.*

As the upper tropospheric cyclone begins to pull out toward the northeast on the 16th, the
surface cyclone intensifies in the region of large $V_{pm}$ south of the Gulf Stream, while the more
gradual warming of the surface air in the region of small $V_{pm}$ north of the Stream yields surface
pressure falls, but not as concentrated and intense as in the CYCLOP to the south.
Although the evolution of the upper tropospheric cyclone differs in detail from the previously
examined cases, and the sharp gradient of sea surface temperature across the north wall of the
Gulfstream clearly plays a role here, in other respects the development of this subtropical
cyclone resembles that of other CYCLOPs, developing in regions of substantial thermodynamic
potential that result, at least in part, from cooling aloft on synoptic time and space scales.
3.6  A Kona Storm
Hawaiians use the term "Kona Storm" to describe cold-season storms that typically form west of
Hawaii and often bring damaging winds and heavy rain to the islands. The term "Kona"
translates to "leeward", which in this region means the west side of the islands. They may have
been first described in the scientific literature by Daingerfield (1921). Simpson (1952) states that
Kona Storms possess "cold-core characteristics, with winds and rainfall amounts increasing with
distance from the low-pressure center and reaching a maxima at a radius of 200 to 500 mi.
However, with intensification, this cyclone may develop warm-core properties, with rainfall and
wind profiles bearing a marked resemblance to those of the tropical cyclone." In general,
Simpson's descriptions of the later stages of some Kona Storms are consistent with them being
CYCLOPs.  But it should be noted that the term is routinely applied to cold-season storms that
bring hazardous conditions to Hawaii regardless of whether they have developed warm cores. A
good review of the climatology of Kona storms is provided by Otkin and Martin (2004), who state
that "development of the surface cyclone in all types results from the intrusion of an upper-level
disturbance of extratropical origin into the subtropics". Here we focus on  Kona storms that do
develop warm cores, providing as a first  example the Kona Storm of 19 December 2010.  A
visible satellite image of this storm is shown in Figure 20.

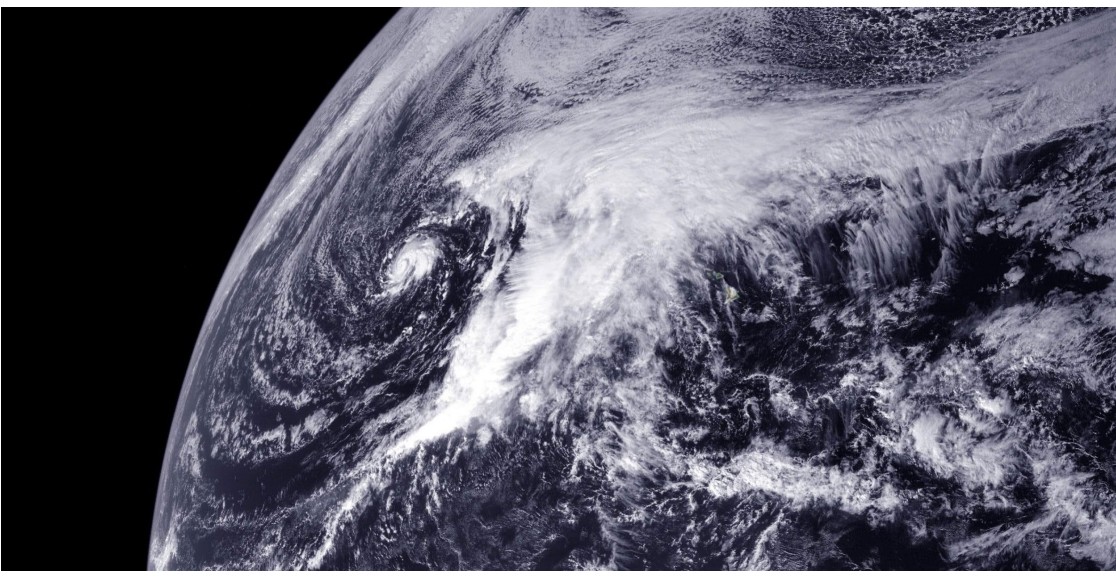


*Figure 20:  Geostationary satellite visible image showing a Kona Storm at 00 UTC on 19 December 2010. The Kona*
*Storm is the small-scale cyclone left of the major cloud mass. (Image credit: NOAA-NASA GOES Project, 2010.)*

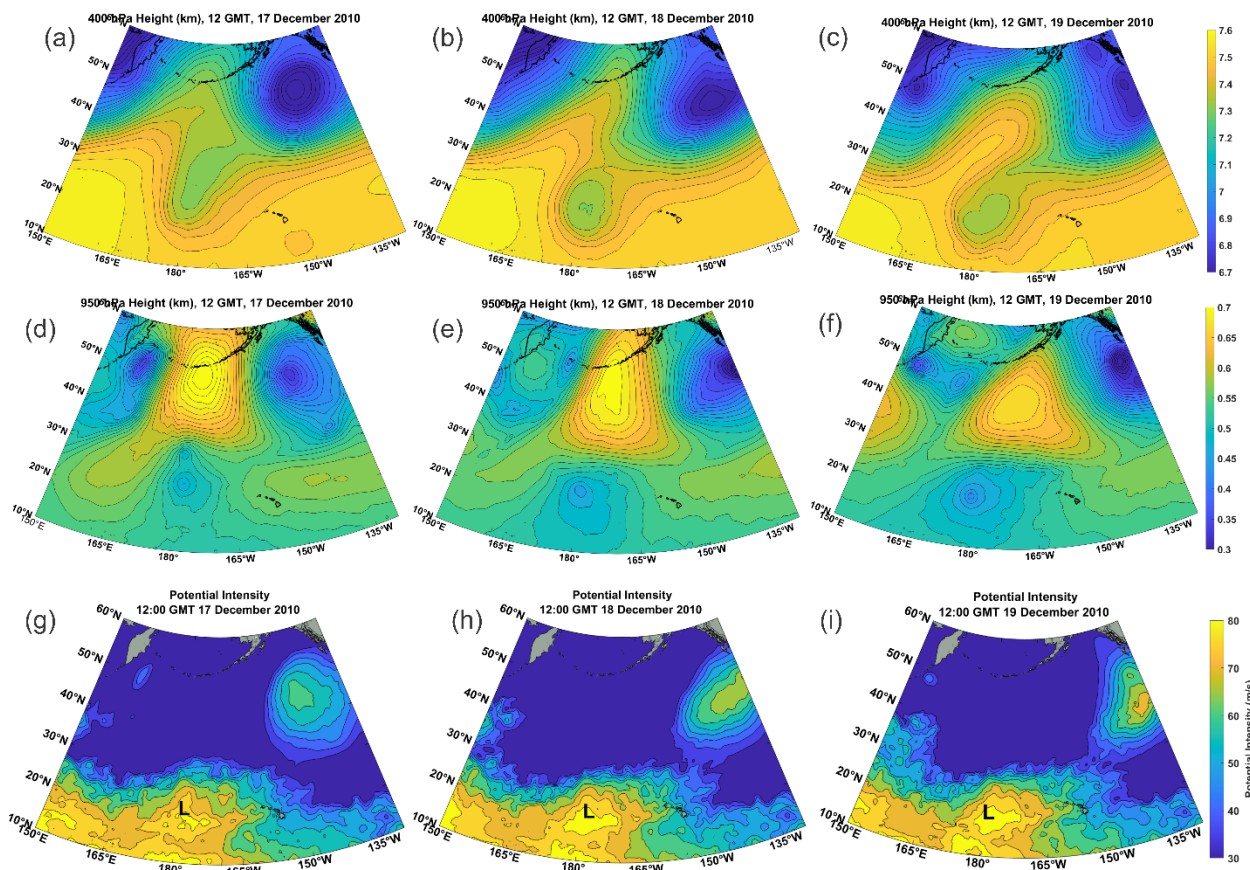


*Figure 21: Sequence of 400 hPa geopotential height charts (km) at contour intervals of 30 m (a)-(c)), 950 hPa geopotential heights (km) with contour intervals of 13.34 m (d)-(f)), and $V_{pm}$ with contour intervals of 5 ms$^{-1}$ ((g)-(i)) at 12:00 UTC on December 17$^{th}$ ((a), (d), (g)), December 18$^{th}$ ((b), (e), (h), and December 19$^{th}$ ((c), (f), (i)) 2010. The "L"s in ((g)-(i)) denote the positions of the 950 hPa cyclone center. From ERA-5 reanalysis.*

As with all known CYCLOPs, the December 2010 Kona Storm developed under a cold-core
cutoff cyclone aloft, as shown in Figure 21. The upper-level cyclone had a long and illustrious
history before December 18$^{th}$, having meandered over a large swath of the central North Pacific.
But beginning on December 17$^{th}$, the cutoff cyclone made a decisive swing southward over
waters with higher values of $V_{pm}$. A broad surface cyclone was present underneath the cold
pool aloft on all three days, but developed a tight inner core on the 19$^{th}$ as the cold pool slowly
drifted over a region of higher potential intensity.
This Kona Storm developed in a region of modest climatological potential intensity that was,
however, substantially enhanced by the cutoff cyclone aloft. For example, on December 17$^{th}$,
the conventional (unmodified) potential intensity at the position of the 950 hPa cyclone center
was about 55 ms$^{-1}$, compared to the 75-80 ms$^{-1}$ values of the modified potential intensity.  One
can only speculate whether a surface flux-driven cyclone would have developed without the
enhanced cooling associated with the cutoff cyclone aloft.
This Kona storm was identified as a tropical storm by the Central Pacific Hurricane Center, as of
0900 UTC 20 December, and given the name Omeka.
The Kona Storm of March, 1951 is a more unambiguous case of a CYCLOP. Figure 22 shows
the 950 hPa geopotential height field on March 26, 1951. The storm, at that time, was classified
by the Joint Typhoon Warning Center as a tropical cyclone, but the climatological potential
intensity there at that time of year could not have supported any tropical cyclone. Figure 22c
shows that substantial $V_{pm}$ was associated with a cutoff cyclone in the upper troposphere,
making possible the existence of this CYCLOP.

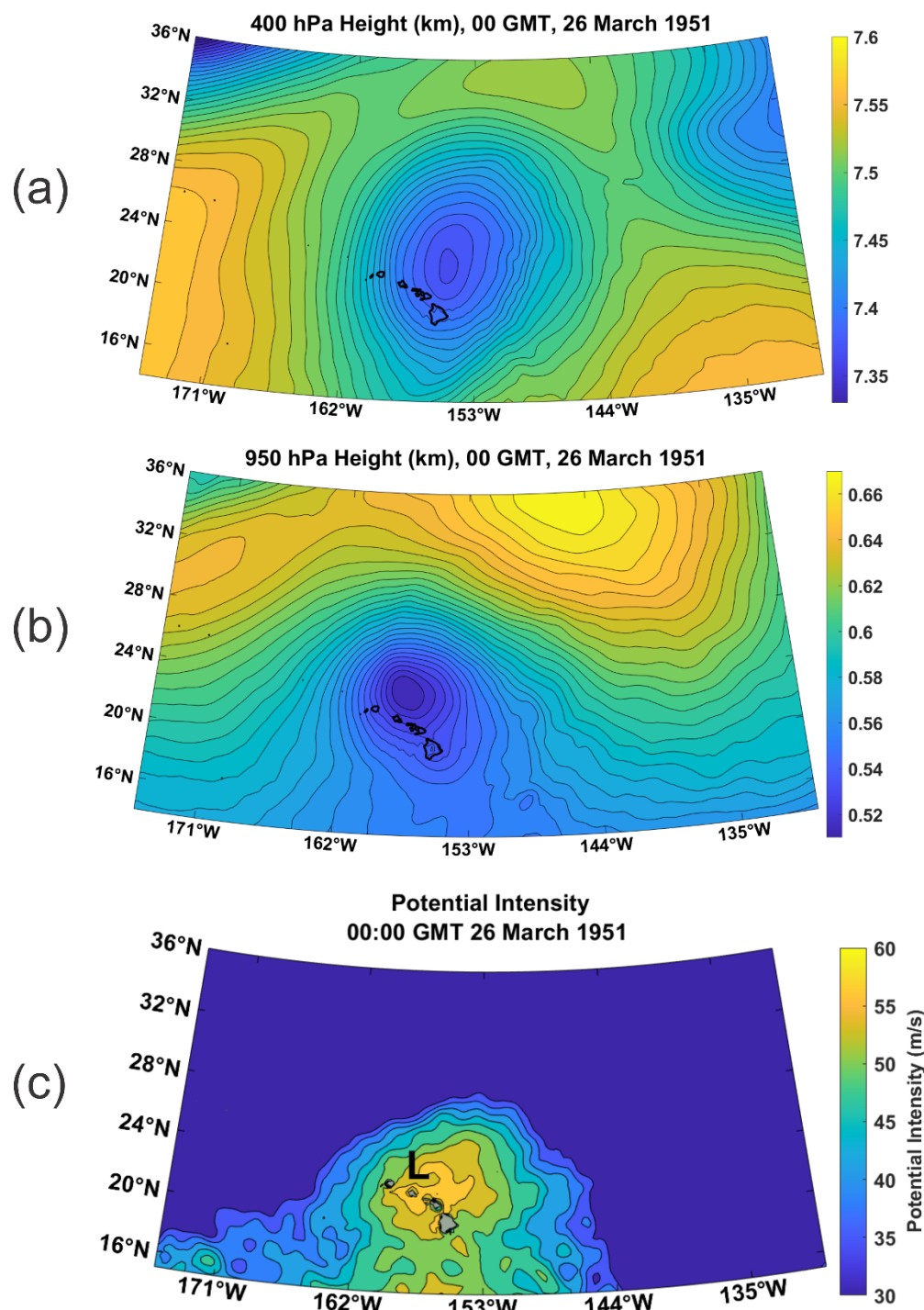


*Figure 22: Kona Cyclone of March, 1951. Fields are shown at 00 UTC on 26 March: a): 400 hPa geopotential height*
*(km)with a 10 m contour interval, b): 950 hPa geopotential height (km) with 5m contour interval, and c): $V_{pm}$ with a*
*contour interval of 3 ms$^{-1}$.*

## 4. Variations on the Theme

We here are attempting to distinguish a class of cyclones, CYCLOPs, from other cyclonic storms by the physics of their development, not by the regions in which they develop. Here we present a case of an actual tropical cyclone in the Mediterranean that we do not identify as a CYCLOP.

### 4.1 Cyclone Zorbas

The cyclone known as Zorbas developed just north of Libya on 27 September 2018 and moved northward and then northeastward across the Peloponnese and the Aegean (Figure 23), dissipating in early October. The storm killed several people and did millions of dollars of damage.

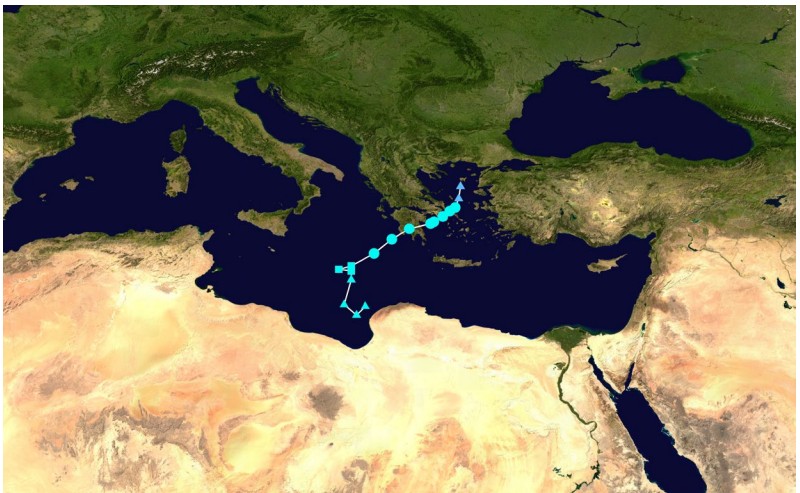

*Figure 23:  Track of Cyclone Zorbas, from 27 September through 2 October, 2018. (Image credit: Wikipedia Commons; https://commons.wikimedia.org/wiki/File:Zorbas_2018_track.png )*

The antecedent (actual, not modified) potential intensity distribution, on 26 September, is displayed in Figure 24. In much of the southern part of the central Mediterranean, the potential intensity was typical of tropical warm pools with values approaching 80 ms$^{-1}$. As with most medicanes, Zorbas was triggered by an upper tropospheric Rossby wave breaking event (Figure 25), but in this case the cold pool aloft only enhanced the existing potential intensity by about 7 ms$^{-1}$ (on September 27$^{th}$). In this case, the maximum 400 hPa geopotential perturbation was around 1500 m$^2$s$^{-2}$, compared to about 2500 m$^2$s$^{-2}$ in the case of Medicane Zeo of 2005. Zorbas was therefore more like  a classic case of a tropical cyclone resulting from tropical transition (Bosart and Bartlo, 1991; Davis and Bosart, 2003, 2004)  and we might  not describe it as a CYCLOP. Note that while the approach of the upper-level cyclone did not appreciably alter the potential intensity, it almost certainly humidified the middle troposphere, making genesis somewhat more likely, as may be true in most cases of tropical transition.

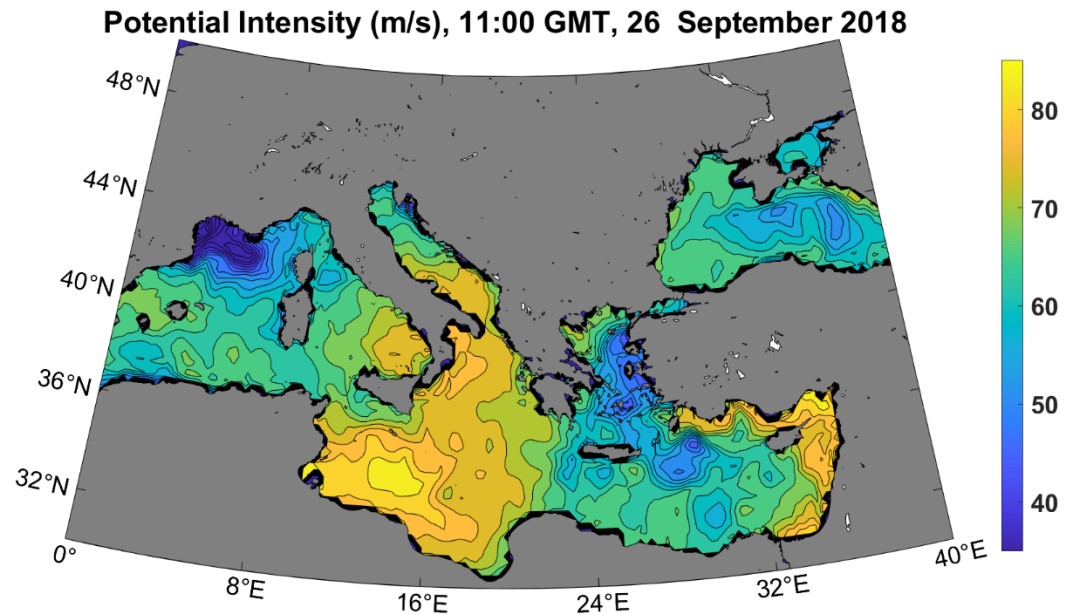

**Potential Intensity (m/s), 11:00 GMT, 26 September 2018**


*Figure 24: Actual potential intensity distribution in the Mediterranean and Black Seas, 11 UTC on 26 September*
*2018.*

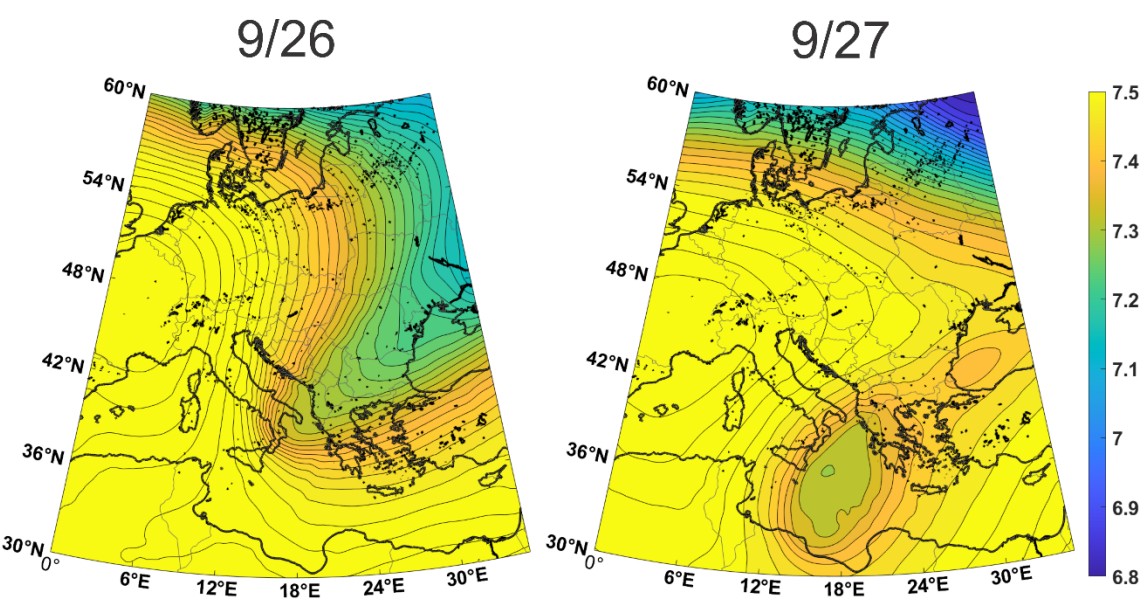


*Figure 25: 400 hPa geopotential height (km) at 11 UTC on 26 September (left) and 27 September (right), 2018.*
*Contour interval is 13.34 m. From ERA-5 reanalysis.*
## 4.2  Cyclone Daniel
Cyclone Daniel of 2023 was the deadliest Mediterranean surface flux-driven cyclone in recorded
history, with a death toll exceeding 4,000 and thousands of missing persons, mostly owing to
floods caused by the catastrophic failure of two dams near Derna, Libya (Flaounas et al., 2024).
This flooding was the worst in the recorded history of the African continent. Figure 26 shows the
track of Daniel's center and a visible satellite image of the storm as it approached landfall in
Libya is shown in Figure 27. Daniel became a surface flux-driven cyclone off the west coast of
the Peloponnese on September 5th, made landfall near Benghazi, Libya, on September 10th and
dissipated over Egypt on the 12th.

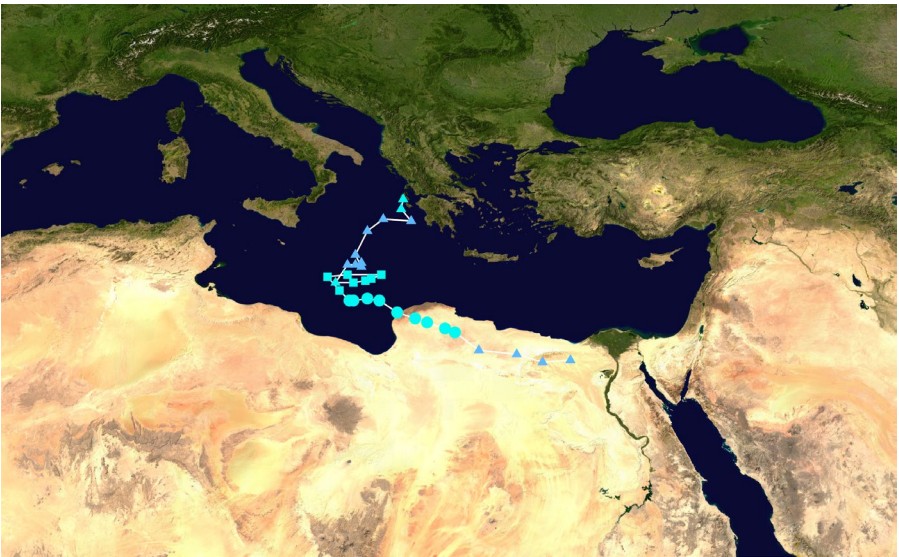


*Figure 26: Track of Storm Daniel at 6-hour intervals, beginning September 5th and ending September 12th, 2023. The*
*circles, squares and triangles along the track denote tropical cyclone, subtropical cyclone and extratropical cyclone*
*designations, respectively, while the deep blue and light blue colors denote tropical depression- and tropical storm-*
*force winds. (Image credit: Wikipedia commons; https://commons.wikimedia.org/wiki/File:Daniel_2023_track.png )*

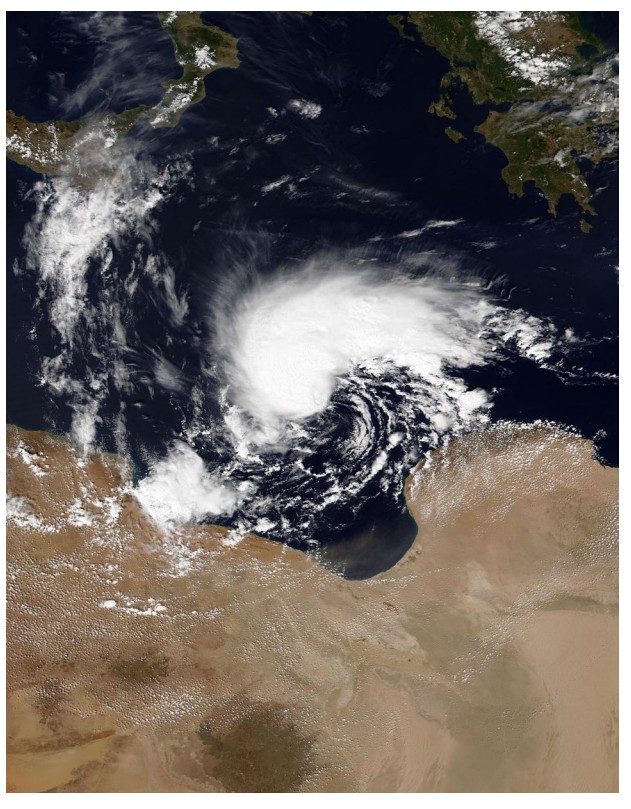

*Figure 27: NOAA-20 VIIRS image of Storm Daniel at 12 UTC 9 September 2023, as it approached the Libyan coast.*


As with Zorbas, the antecedent potential intensity was high throughout the Mediterranean south
and east of Italy and Sicily, and the event was triggered by a Rossby wave breaking event
(Figure 28). And as with Zorbas, the cutoff cyclone aloft was not strong enough to appreciably
alter the existing potential intensity but acted as a trigger for the tropical cyclone that Daniel
became. This was another classic case of tropical transition, and not a CYCLOP.

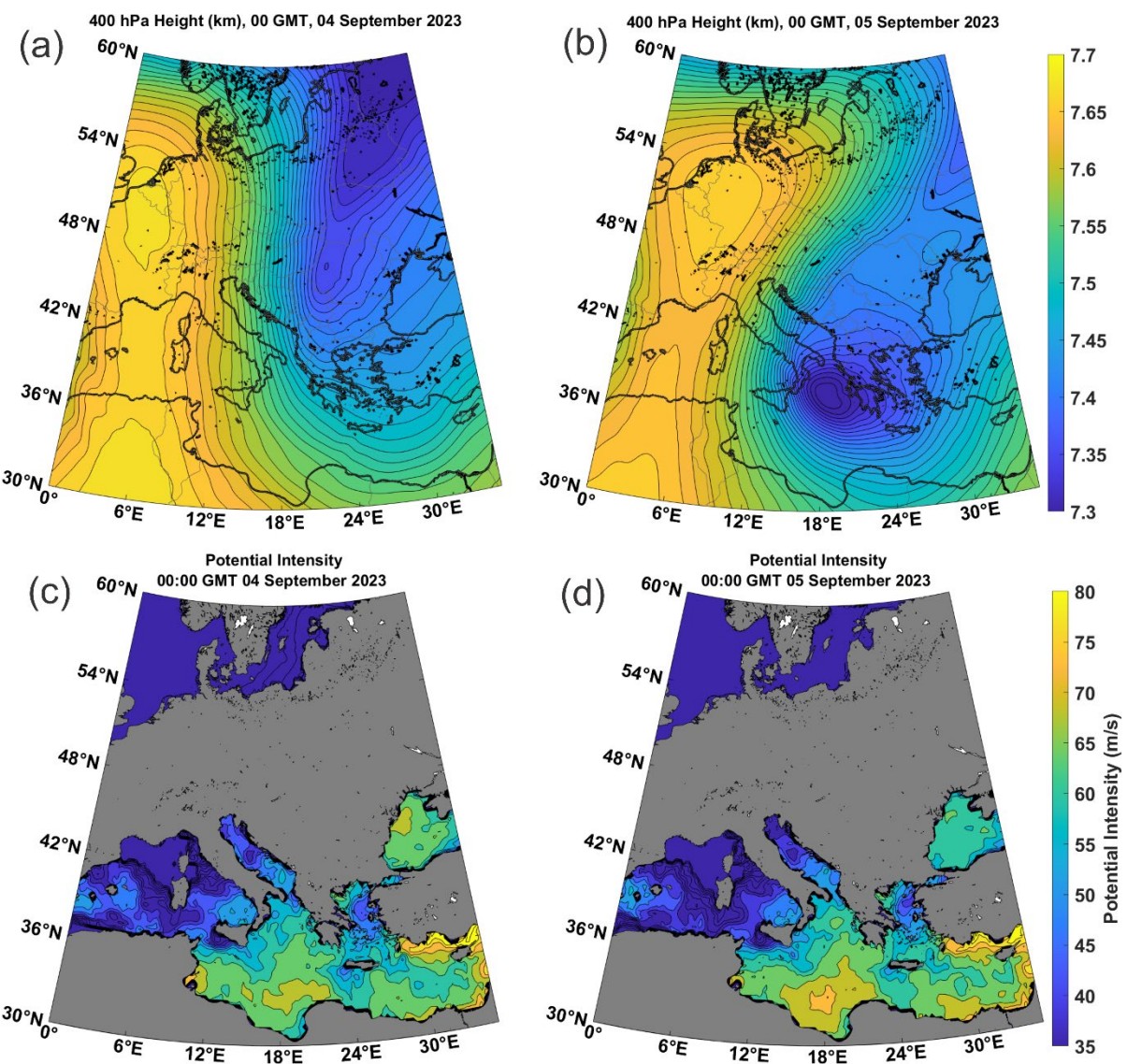


*Figure 28: 400 hPa geopotential height (km) with contour interval of 13.33 m (a)-(b); and actual potential intensity (ms⁻¹) with contour interval of 4 ms⁻¹ (c)-(d) at 00 UTC September 4th ((a) and (c)) and 5th ((b) and (d)), from ERA-5 reanalyses.*

Yet Storm Daniel differed from Zorbas in one important respect: as it approached the Libyan coast around September 10th, it came under the influence of strong high-level potential vorticity (PV) advection owing to a mesoscale "satellite" PV mass rotating around the upper-level cutoff cyclone (Figure 29). The quasi-balanced forcing associated with the superposition of the high-level PV anomaly with the surface-based warm core probably contributed to Daniel's intensification which, remarkably for a surface flux-driven cyclone, continued after landfall (Hewson et al., 2024).

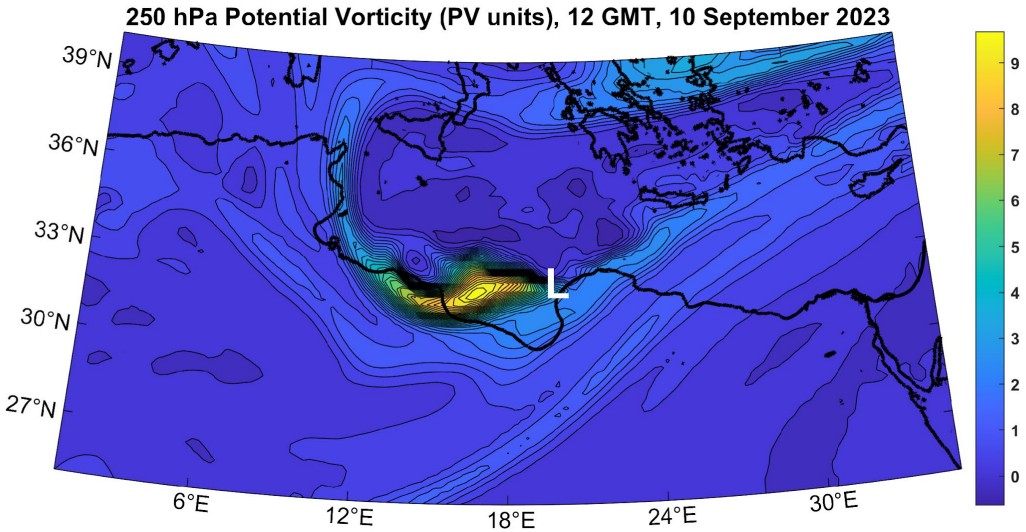

**250 hPa Potential Vorticity (PV units), 12 GMT, 10 September 2023**


*Figure 29: Potential vorticity (PV units, $10^{-6}$ K $kg^{-1}$ $m^2$ $s^{-1}$) at 250 hPa at 12 UTC on 10 September, 2023. The white*
*"L" marks the approximate surface center of Daniel at this time.*

## 5. Summary and conclusions

We here argue that many of the cyclonic storms called medicanes, polar lows, subtropical
cyclones, and Kona storms operate on the same physics and ought to be identified as a single
class of storms that we propose to call CYCLOPs. In their developed structure and intensity
they are essentially identical to tropical cyclones, and like TCs are driven primarily by wind-
dependent surface enthalpy (latent and sensible heat) fluxes, but unlike classical TCs, there is
little or no climatological potential intensity for the storms. Rather, the development and/or
migration over relatively warm water of strong, cold-core cyclones in the upper troposphere
cools and moistens the column through dynamical lifting, generating mesoscale to synoptic
scale columns with elevated potential intensity and humidity, and reduced wind shear – ideal
embryos for the development of surface flux-driven cyclones.
We do not expect CYCLOPs to last as long as classical TCs. In the first place, the conditions
that enable such storms are confined in space and transient in time. For example, the cutoff
cyclone aloft is often re-absorbed into the main baroclinic flow. In addition, the strong surface
enthalpy fluxes that power CYCLOPs also increase, through the action of deep convection, the
enthalpy of the otherwise spatially limited cold columns in which they form, reducing over time
the thermodynamic disequilibrium between the air column and the sea surface. A back-of-the-
envelope estimate for the time to destroy the initial thermodynamic disequilibrium is on the order
of days. By contrast, even the strongest classical TCs do not sufficiently warm the large
expanses of the tropical troposphere they influence to appreciably diminish the large-scale
potential intensity of tropical warm pools.
Not all cyclonic storms that have been called polar lows, medicanes, subtropical cyclones, or
Kona storms meet our definition of CYCLOP. The literature describes many polar lows that have
been traced to something more nearly like classical baroclinic instability acting in an air mass of
anomalously low static stability (e.g. Sardie and Warner, 1985). Many storms identified as Kona
storms because of their location and season, and which developed under cold cyclones aloft,
never received much of a boost from surface fluxes and therefore would not be classified as
CYCLOPs. And, as we described in the last section, two strong Mediterranean cyclones, Zorbas
of 2018 and Daniel of 2023, developed in environments of high climatological potential intensity
and formed via the tropical transition process.
Beyond these caveats, not all cold-core, closed cyclones in the upper troposphere that develop
or move over relatively warm ocean waters develop CYCLOPs. Without doing a comprehensive
survey, we have found several cases of greatly enhanced potential intensity under cold lows
aloft that did not develop strong, concentrated surface cyclones. We suspect that, as with
classical tropical cyclones, dry air incursions above the boundary layer may have prevented
genesis in these cases, but as these events generally occur in regions devoid of in-situ
observations, it is not clear how well reanalyses capture variations of moisture on the scale of
CYCLOPs. In any event, the efficiency with which upper-level cut-off cyclones produce
CYCLOPs, given a substantial perturbation in potential intensity, should be a subject of future
research.
Clearly, there exists a continuum between pure tropical transition, in which synoptic-scale
dynamics play no role in setting up the potential intensity, and pure CYCLOPs in which
synoptic-scale processes create all the potential intensity that drives the storm.  The distinction
we are making here does not pertain to the final result – a tropical cyclone[6] – but to the route to
creating it.  As a practical matter,  forecasters  should consider both the triggering potential and
mesoscale to synoptic scale environmental development in predicting the formation and
evolution of CYCLOPs. The modified potential intensity ($V_{pm}$) introduced here may prove to be
a valuable diagnostic that can easily be calculated from NWP model output. To simulate
CYCLOPs, NWP models need to make accurate forecasts of dynamic processes that lead to
the formation and humidification of deep cold pools aloft, and be able to handle surface fluxes
and other boundary layer processes essential to the formation of surface flux-driven cyclones.
And, as with tropical cyclones, coupling to the ocean is essential for accurate intensity
prediction.
Finally, we encourage researchers to focus on the essential physics of CYCLOP development
regardless of where in the world they occur. Casting a broader geographical net will harvest a
greater sample of such storms and should lead to more rapid progress in understanding and
forecasting them.

---

[6] It might be beneficial to develop a term that generically denotes a cyclone driven primarily by surface
enthalpy fluxes. "Tropical cyclone" seems unsuited to the task, since "tropical" hardly describes, e.g., a
polar low. Such a generic term would encompass both conventional tropical cyclones, which occur in
regions of climatologically large potential intensity, and CYCLOPs, for which potential intensity is
generated locally in space and time.

# Appendix


CYCLOPs form when dynamical processes create synoptic-scale cold columns marked by
cutoff cyclones in the upper troposphere. The objective here is to calculate tropical cyclone
potential intensity in these cold columns. The problem is that the CYCLOPs themselves, and
surface heat fluxes in general, warm the columns, sometimes rapidly, diminishing the potential
intensity. We want to know what the potential intensity was before this warming occurs. Here we
develop a modified potential intensity using the surface pressure depression as proxy for the
column heating.
Potential intensity is a measure of the maximum surface wind speed that can be achieved by a
cyclone fueled entirely by surface enthalpy fluxes. It is defined (see Rousseau-Rizzi and
Emanuel (2019) for an up-to-date definition):
$$V_p^2 = \frac{C_k}{C_D} \frac{T_s - T_o}{T_o} \left( h_0^* - h_b \right), \tag{A1}$$

Where $C_k$ and $C_D$ are the surface exchange coefficients for enthalpy and momentum, $T_s$ and
$T_o$ are the absolute temperatures of the surface and outflow layer, $h_0^*$ is the saturation moist
static energy of the sea surface, and $h_b$ is the moist static energy of the boundary layer.
Using the relation $T\delta s \simeq \delta h$, where $s$ is moist entropy, we can re-write (A1) slightly as
$$V_p^2 = \frac{C_k}{C_D} \frac{T_s - T_o}{T_o} T_s \left( s_0^* - s_b \right). \tag{A2}$$

Next, we assume that the troposphere near CYCLOPs has a nearly moist adiabatic lapse rate[7],
and the moist entropy of the boundary layer is equal to the saturation moist entropy, $s^*$, of the
troposphere; i.e., that the troposphere is neutrally stable to moist adiabatic ascent from the
boundary layer. Under these conditions, (A2) may be re-written
$$V_p^2 = \frac{C_k}{C_D} \frac{T_s - T_o}{T_o} T_s \left( s_0^* - s^* \right). \tag{A3}$$

Here $s^*$ is constant with altitude, since the troposphere is assumed to have a moist adiabatic
lapse rate.

---

[7] If the lapse rate is appreciably sub-adiabatic, then convection cannot occur and no flux-driven cyclone could develop. This condition would be associated with a larger value of s* in (A3), which would at least reduce the calculated value of $V_p$. Super-adiabatic lapse rates are not generally observed over the ocean within deep humid layers.

Referring to Figure 1a in the main text, we want to know what the potential intensity is in the
vicinity of the cutoff cyclone aloft *before* the atmosphere underneath it has started to warm up
under the influence of surface enthalpy fluxes.  But in reality, this warming commences as soon
as the system moves over relatively warm water and deep convection begins. We can estimate
what the temperature, or $s^*$, was before the warming began by using the surface pressure drop
as a proxy for the column warming.
We begin with the perturbation hydrostatic equation in pressure coordinates:

$$\frac{\partial \phi'}{\partial p} = -\alpha',$$

(A4)
where $\phi'$ is the perturbation geopotential and $\alpha'$ is the perturbation specific volume. Since the
latter is a function of pressure and $s^*$ only, we can use the chain rule and one of the Maxwell
relations from thermodynamics (Emanuel, 1994) to write (A4) as

$$\frac{\partial \phi'}{\partial p} = -\left(\frac{\partial T}{\partial p}\right)_{s^*} s^{*\prime}.$$

(A5)
Now, since $s^{*\prime}$ does not vary with altitude, we can integrate (A5) from the surface to the local
tropopause to yield

$$-\phi_s' = \left(T_s - T_{\phi'=0}\right) s^{*\prime}.$$

(A6)
where $\phi_s'$ is the near-surface geopotential perturbation and $T_{\phi'=0}$ is the absolute temperature at
the level where the CYCLOP-induced geopotential perturbation vanishes.
Therefore the CYCLOP-induced warming of the troposphere is estimated as

$$s^{*\prime} = \frac{-\phi_s'}{T_s - T_{\phi'=0}}.$$

(A7)
We subtract this warming from the reanalysis environmental saturation entropy in (A3), yielding

$$V_{pm}^2 = \frac{C_k}{C_D} \frac{T_s - T_o}{T_o} T_s \left(s_0^* - s_e^* - \frac{\phi_s'}{T_s - T_{\phi'=0}}\right),$$

(A8)
where $s_e^*$ is the analyzed saturation entropy of the column. This can be written alternatively as

$$V_{pm}^2 = V_p^2 - \frac{C_k}{C_D} \frac{T_s}{T_o} \frac{T_s - T_o}{T_s - T_{\phi'=0}} \phi_s',$$

(A9)
where $V_p$ is the potential intensity calculated from the reanalysis. We use (A9) in the
calculations reported in this paper, with $V_p$ calculated using the algorithm of Bister and Emanuel

743      (2002).

For the purposes of the present work, we defined the perturbation as the difference between the
actual geopotential and its climatological value determined from monthly mean values over the
period 1979-2023, and we estimate the near-surface cyclone geopotential perturbation as that
at 950 hPa. In (A9), we do not know the precise value of $C_k / C_D$, but it should be of order unity
and is generally assumed as such when potential intensity is calculated in practice (e.g.
Emanuel, 2000). The surface temperature, $T_s$, should be captured well in reanalyses, while the
outflow temperature is often taken as the local unperturbed tropopause temperature (Emanuel,
2000). In general, the pressure perturbation associated with tropical cyclones changes sign
somewhere in the upper troposphere, and for simplicity we take $T_{\phi'=0} = T_o$ here.      In the tropics,
the surface temperature can be around 300 K while the tropopause is often as cold as 200K,
yielding $T_s / T_o = 1.5$, while in polar latitudes, the surface temperature can be just above freezing
(273 K) while the tropopause is warmer, around 220 K, giving $T_s / T_o = 1.25$. As in any case
$C_k / C_D$ is uncertain, we take $T_s / T_o = 1.3$ here for simplicity, and approximate (A9) as
$$V_{pm}^2 = V_p^2 - 1.3\,\phi_s'. \tag{A10}$$

The contribution of the last term in (A10) is usually small except in the immediate vicinity of a
CYCLOP.
The individual contributions of the two terms on the right side of (A10) are displayed in Figure
A1, which pertains to the case of Medicane Celeno on the 14th and 15th of January, 1995. The
total (modified) potential intensity for these two dates and times is shown in Figure 5 *g* and *h*. In
general, the contribution of the correction (last term in (A10)) is modest.
Code and data availability
No modeling was performed in the course of this work, and no code developed except to plot
reanalysis data. Routines for calculating potential intensity are available at
https://github.com/dgilford/tcpyPI . All of the meteorological analyses presented herein are
based in ERA5 downloaded from the Copernicus Climate Change Service
(https://doi.org/10.24381/cds.143582cf, Hersbach et al., 2020).
Author contributions
KE carried out the analyses and prepared the manuscript with contributions from all the co-
authors.

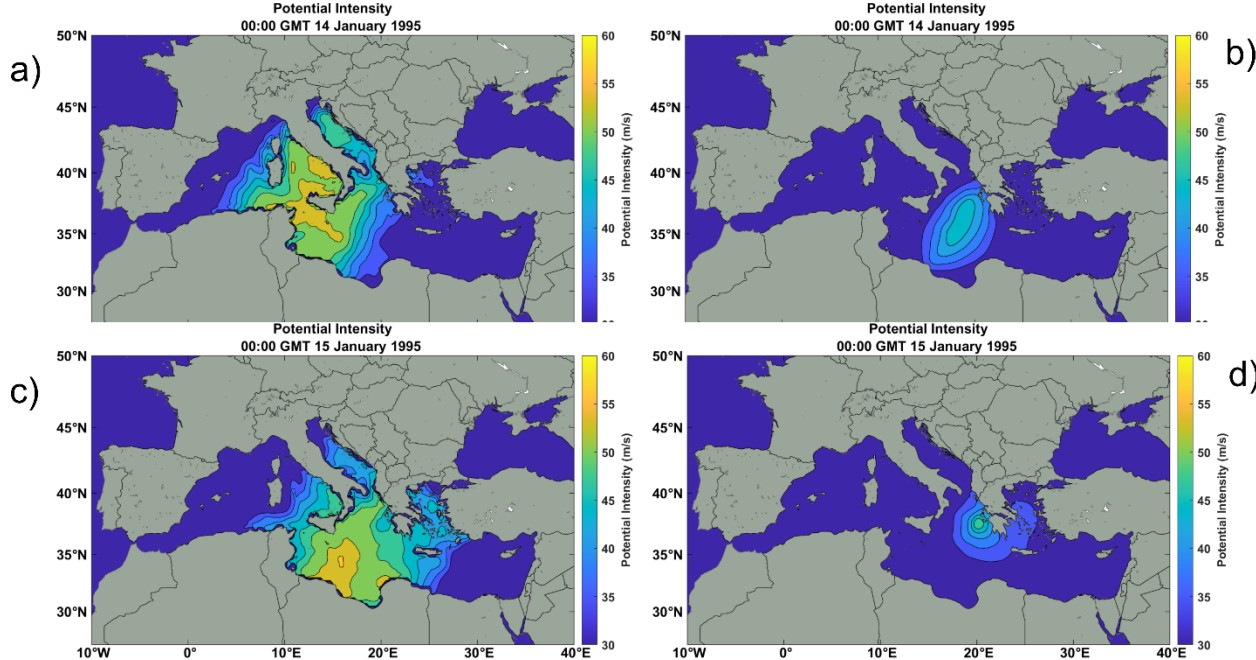

*Figure A1: Unmodified potential intensity (a and c) and modification of the potential intensity (b and d) at 00 UTC on*
*January 14th (a and b) and 15th (c and d). Note that the full modified potential intensity is not the arithmetic sum of*
*these two contributions but rather their Euclidean norm.*
## Competing interests
The authors declare that they have no conflict of interest.
## Acknowledgements
We thank two anonymous reviewers for most helpful and conscientious reviews, and Lance
Bosart for his detailed and highly constructive comments.
This paper was motivated by discussions at the TROPICANA (TROPIcal Cyclones in
ANthropocene: physics, simulations & Attribution) program, which took place in June 2024 at
the Institute Pascal, University of Paris-Saclay, and aimed to address complex issues related to
tropical cyclones, medicanes, and their connection with climate change.
This work was made possible by Institut Pascal at Université Paris-Saclay with the support of
the program TROPICANA, "Investissements d'avenir" ANR-11-IDEX-0003-01.
T. Alberti and D. Faranda acknowledge useful discussions within the MedCyclones COST
Action (CA19109) and the FutureMed COST Action (CA22162) communities.
S. Bourdin received financial support from the NERC-NSF research grant n° NE/W009587/1
(NERC) & AGS-2244917 (NSF) HUrricane Risk Amplification and Changing North Atlantic
Natural disasters (Huracan), and from the EUR IPSL-Climate Graduate School through the
ICOCYCLONES2 project, managed by the ANR under the "Investissements d'avenir"
programme with the reference 37 ANR-11-IDEX-0004 - 17-EURE-0006.
S.J. Camargo and C.-Y. Lee acknowledge the support of the U.S. National Science Foundation
(AGS  20-43142, 22-17618, 22-44918) and U.S. Department of Energy (DOE) (DE-
SC0023333).
D. Faranda, E. Flaounas were supported by the COST Action CA22162 - FutureMed: A
TRANSDISCIPLINARY NETWORK TO BRIDGE CLIMATE SCIENCE AND IMPACTS ON
SOCIETY (FutureMed).
K. Emanuel was supported by the U.S. National Science Foundation under grant AGS-2202785.
J.J. González-Alemán thanks support from AEMET and from the Spanish PID2023-146344OB-
I00 project, funded by MICIU/AEI/10.13039/501100011033 y by FEDER, EU
M. M. Miglietta was partly supported by "Earth Observations as a cornerstone to the
understanding and prediction of tropical like cyclone risk in the Mediterranean (MEDICANES)",
ESA Contract No. 4000144111/23/I-KE.
H. Ramsay acknowledges funding support from the Australian Climate Service and the Climate
Systems Hub of the Australian Government's National Environmental Science Program (NESP).
M. Reale was supported by the National Recovery and Resilience Plan project TeRABIT
(Terabit network for Research and Academic Big data in Italy - IR0000022 - PNRR Missione 4,
Componente 2, Investimento 3.1 CUP I53C21000370006) in the frame of the European Union -
NextGenerationEU funding.
R. Romero acknowledges financial support by the "Ministerio de Ciencia e Innovación" of Spain
through the grant TRAMPAS (PID2020-113036RB-I00/AEI/10.13039/501100011033).

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
