# Peer review of "Kerry Emanuel (1), Tommaso Alberti (2), Stella Bourdin (3), Suzana J. Camargo (4,5), Davide"

_EGUsphere, 2024_

## Author Response (AR1)

Responses to the Reviewers of

**A Unified Framework for Surface Flux-Driven Cyclones Outside the Tropics**

**Reviewer 1**

We thank Reviewer 1 for his or her exceptionally conscientious and comprehensive review, which we found to be of great value in improving our paper. Herewith our responses to the reviewer's points. Quotations from this review are italicized in what follows.

*Although the concept of CYCLOPs is an innovative and useful heuristics, however, their deepening is mostly driven by a "spatio-temporally localized wind-induced surface heat exchange" (WISHE) and this makes them much more akin to tropical cyclones than to extratropical ones (driven by baroclinic instability). From this point of view, they would not deserve their own equidistant "corner" in the cyclone space (Fig. 30), and they could be more easily assimilated to the "trough-supported" tropical transition cases over cold water discussed by McTaggart-Cowan et al. (2015). Explaining the several unexpected tropical transitions observed over "cold" waters using the CYCLOP concept is already a very valuable contribution that this research work brings to the literature.*

*I believe that part of the problem lays in a misunderstanding between processes setting up the preconditioning of the storm environment and the actual energetic driving of cyclones: while tropical cyclones are usually distinguished from extratropical cyclones through their energy source driving their deepening (i.e., WISHE vs baroclinic instability), in this paper CYCLOPs are distinguished from "classic" tropical cyclones through a difference in their preconditioning (the availability-or not- of PI in the environment where the low develops). This fundamental difference is not always clearly expressed in the manuscript, leading to some ambiguities that I tried to pinpoint in the comments below.*

We agree with the reviewer. Throughout the revised paper, we have removed prose that implied that CYCLOPs are a different kind of cyclone and inserted text that emphasizes the difference in the route to formation. We have eliminated Figure 30.

*The other critique that could be moved to the paper is that it does not go the final mile in emphasizing the broad implications of the performed analysis. The title "A Unified Framework for Surface Flux-Driven Cyclones Outside the Tropics" suggests the attempt to "unify" all storms driven by "surface fluxes" (or to be more precise, by the feedback between surface fluxes, deep convection and wind speed conceptualized by WISHE), but the outcome of the paper is a rather divisive operation, that attempts to separate the CYCLOP category from supposedly more "pure" surface-flux driven cyclones (as in Sec.4, but also Sec. 5 and 1). The presence or the absence of PI in the environment is very interesting to discuss in order to understand "special" cases such as medicanes or polar lows, but does not result in storms with fundamentally different properties than tropical cyclones: the final result of the process is still a -more or less transient- warm core, surface-flux driven storm. On the other hand, this piece of work provides a powerful and still unexploited unifying platform, because the extension of PI concept to the extratropics is shown to encompass virtually every known category of surface-flux driven*

*cyclones, even allowing to assess whether a given storm can or cannot be surface-flux driven (e.g., at ll. 612-615).*

We agree that fully developed CYCLOPs are isomorphic to tropical cyclones and emphasize that the difference lies in the local development of potential intensity. In the revised paper we emphasize this distinction and suggest that forecasters use potential intensity more routinely as a diagnostic, even outside the tropics. We also propose that the community develop a new term that encompasses both conventional tropical cyclones and surface flux-driven cyclones outside the tropics.

Responses to detailed comments:

1. *Can cyclones be "partly driven" by extratropical or tropical energy sources (ll. 58-66)? To answer this question, one has to define the meaning of "driving", the way to measure "partly" and which physical processes are involved and termed "tropical" and "extratropical". As CYCLOPS lay at the intersection of many meteorological concepts, it is important to address those conceptual challenges before dwelling deeper. First of all, the driving. Both tropical and extratropical cyclones can develop only if the large-scale flow configuration favors (or, more strictly, allows) their development: extratropical cyclones are not observed under upper-level ridges, for instance, and tropical cyclones require an initial disturbance and low tropospheric vertical wind shear/dryness to spin up. Once such conditions come in place, the deepening of the cyclonic disturbance can be driven first-order by baroclinic instability or by WISHE, the classical "extratropical" and "tropical" energy sources for cyclones, respectively. Here comes the distinction between preconditioning and driving. the former is required so that instabilities can be unleashed and "drive" a negative sea-level pressure tendency in the cyclone. In the case of CYCLOPs, the large-scale flow serendipitously creates mesoscale "pockets" where surface flux-driven instability is allowed, and that are exploited by CYCLOPs to spin up (as discussed by the authors, e.g., at ll.595-599): here the large-scale extratropical flow sets the preconditioning, but the energy source driving cyclone's development or preventing it from decay is fully the "tropical" one, i.e., the driving by surface enthalpy fluxes of the WISHE paradigm. The same distinction can be applied to the example of TC development embedded in African easterly waves, cited at ll. 64-66. Extratropical transitioning storms are, on the other hand, an example of the opposite pathway: the large-scale flow creates favorable conditions for baroclinic instability to occur, and baroclinic development occurs as an upper-level trough "phase-locks" with the warm, moist air mass constituted by the TC. In the ET context, the TC is part of the preconditioning like the upper-level cold-core cyclones leading to CYCLOPS (even if the tropical origin might result in special features such as enhanced precipitation and latent heat release), but the energy source driving cyclone's deepening during ET is baroclinic instability (i.e., the "extratropical" one) often with the synergistic interaction of latent heat release but clearly outside the WISHE framework. The case of storm Daniel in Fig. 29 is quite special, and reminiscent of TC-trough interactions (e.g., Fischer et al. 2019), but likely not the norm. Keeping these distinctions in mind, writing that cyclones of synoptic and sub-synoptic scale can be powered by "one or both" of the two energy sources (l. 58) might be misleading, as the reader would be tempted to mix preconditioning and driving to imagine CYCLOPS as "hybrid" storms driven by the two energy sources at the*

*same time. The two energy sources actually stand in stark contrast, as they operate in very different environments and result in systems with fundamentally different spatial scales and properties. To clear up any ambiguity, I would ask the authors to remove the words "or both" from line 58 and "usually" at line 63, and screen the paragraph -and the rest of the paper- to make clear that "hybrid" storms stand at the intersection of extratropical preconditioning and tropical driving or vice-versa, and not at the intersection of different drivings.*

We agree that in this particular context, stating that cyclones can be driven by one or both processes (baroclinic instability, WISHE) could be misleading, but we here are making a more general statement about cyclones. Hybrid storms can be driven by both processes at the same time, and we now provide a reference for that possibility. But we do not argue that both processes are at work at the same time in the CYCLOPs cases we examined, though in the December, 2005 medicane, warm advection did serve to deepen the synoptic scale precursor surface cyclone.

2. *Do CYCLOPS "stand alone" with respect to tropical and extratropical cyclones (ll. 591-593, 627-639, Fig. 30)? CYCLOPS are depicted as driven by surface fluxes (e.g., ll. 593-594): given that contemporary driving by baroclinic instability is not possible (see point 1 above), I am thus hesitant to give CYCLOPS a status equivalent to tropical and extratropical cyclones, as implicitly suggested in the diagram of Fig. 30. (Again, by "tropical" cyclones it is meant here "purely surface flux-driven cyclones", and by "extratropical" cyclones it is meant "purely baroclinic instability-driven cyclones"). Given their surface-flux driving, the triangle in Fig. 30 should at least not be equilateral, but CYCLOPS should be much closer to the tropical cyclone vertex. However, maybe it would be better to change the way CYCLOPs are included in the conceptual diagram because, at the end of the day, they can be fundamentally assimilated to "tropical", surface-flux driven cyclones. This criticism does not exclude that, in the future, diagnostic tools such as the phase space diagram by Hart (2003) might be extended to distinguish CYCLOPS from other surface flux-driven cyclones (e.g., including geopotential anomaly at 400hPa?), but this would require a separate analysis and is rather a technical challenge, as opposed to the conceptual issue raised here. In addition, in the text it is also written that "real storms migrate through that phase space over time" (l. 631), but the path "TC to CYCLOP" has not been discussed and is not easily conceivable. This issue is paired to the yellow arrows and titles along the edges of the triangle, which are suggestive of possible differences in "driving" and are not discussed in the text or in the caption. To avoid possible misunderstanding in the driving energy sources of CYCLOPS, and for reasons of scientific parsimony, I suggest the authors to reconsider their Fig. 30 and its description and consider CYCLOPS as a type of tropical cyclone that can be reached from the blue "baroclinic" corner via tropical transition in a low-PI environment.*

We have indeed reconsidered, and agree with the reviewer on this point. We took away Figure 30.

3. *Does it make sense to separate CYCLOPS from "pure" tropical transition (ll. 260-265, 526,542, 615-617, 627-629)? Tropical transition can be described as the series of "dynamic and thermodynamic transformations required to create a warm-core cyclone*

*from a cold-core one" (paraphrasis from Davis and Bosart 2004). From this point of view, there is no difference between the cases of Medicane Celeo and Zeo (Sec. 3.1, 3.2) and the ones of Zorbas and Daniel (Sec. 4.1, 4.2): in all cases, an initially cold-core vortex led to the genesis of a warm-core one, fulfilling the basic definition of tropical transition. The only difference between the two sets of medicanes is the presence -or not- of PI in the storm environment, and the relative contribution of the upper-level cold vortex in generating it these differences in preconditioning are definitely worth discussing, and the authors do it in a clear way, but are not enough -in the opinion of the writer- to implicitly define a "CYCLOP transition" that were to be fundamentally distinct from a supposedly idealized, "pure" tropical transition. Unless the authors can define and place "pure tropical transition" and "CYCLOP transition" in the context of existing literature (e.g., Davis and Bosart 2003, McTaggart- Cowan et al. 2015) and justify the need of a separate classification, I would ask them to recalibrate their characterization to remove this ambiguity, possibly introducing a unifying high-level "surface-flux-driving transition" category that generates all the "surface-flux driven cyclones" of the title. I would suggest the authors to frame along a different axis (in Sec. 4) what made Zorbas and Daniel special: the fact that substantial amount of PI were already present above the Mediterranean sea in September (that features climatologically the highest SSTs), a signature of the ongoing tropicalization of Mediterranean climate that might lead, in the future, to more frequent and stronger surface flux-driven cyclones in the region. But again, the difference lies in the preconditioning, rather than the surface-flux driving, and this should be made explicit.*

In the revised manuscript, we make it clear that it is the local (in space and time) generation of PI that distinguishes CYCLOPs from other developments, including tropical transition. In our view, this is an important distinction, from both theoretical and practical standpoints. If we could magically remove the locally generated PI, a warm-core, TC-like cyclone would not happen and in that case we would not even make the mistake of mis-identifying the event as tropical transition, since in the end no TC-like entity would form. From a practical point of view, identifying the generation of modified PI in guidance would become an important consideration, which it is certainly not today.

4. *The considerations outlined in the previous three points lead to a further comment: does it make sense to introduce CYCLOPS as a separate category of cyclones from classical "tropical" cyclones (e.g., ll 525-526, 591-599)? The answer can be nuanced. In my opinion, CYCLOPS are a useful heuristics to explain the occurrence of surface-flux driven cyclones in regions or months where such systems should not be expected a priori, filling a gap of understanding that had not been addressed in such a comprehensive way up to now. They might even be useful from a communication point of view. However, it should be madeclear that the difference between CYCLOPS and "pure" tropical cyclones stem only from differences in the background conditions that lead to their formation (the preconditioning) and not in their driving. I guess the authors are already aware of this difference and would agree with this comment, so what I ask is to do even more in that direction. After all, the term "tropical cyclone" is actually an old reminiscence of a period where such flux-driven systems were thought to exist only in the tropical regions: the key result of this study, on the other hand, is that we should move to more inclusive terms like "surface flux-driven cyclones", in the spirit of Emanuel*

*(1988), to encompass "classic" hurricanes and any other low pressure system that, at a given moment of their life cycle, experiences a pressure drop because of wind/surface flux feedbacks. Once this category is defined with respect to the driving (following the classic distinction between tropical and extratropical cyclones), it is possible to start stratifying with respect to the preconditioning and distinguish, e.g., according to the availability of PI in the storm environment and its consequences on the storm evolution. I would suggest to the authors, as actionable point, to further emphasize in Sec. 5 the commonalities between CYCLOPS and "pure TCs" as surface-flux driven cyclones, and not only their differences: the title indeed hints to an attempt to "unify" surface-flux driven cyclones, but I feel this initial aim is not really accomplished in Sec. 5. After all, this work has shown that the genesis of virtually all storms with some degree of tropical characteristics around the world is due to the in situ genesis and/or presence of PI -and even a provides a necessary condition to ascertain surface flux driving on a case-by-case basis!-, in agreement with the WISHE concept, and this is a very interesting result.*

We agree that there is a certain awkwardness in defining an entity (CYCLOP) when the distinction is really in the route to its creation. (To take a slightly humorous analogy, a "bastard" is technically someone who was conceived "illegitimately", but biologically and in all other respects that person is a human being like all other human beings. We could say a CYCLOP is like a bastard.) We do strongly agree that we need a term that encompasses all surface flux-driven cyclones, including TCs, but do not propose such a term here. The transience of the PI pool that supports CYCLOPs is an important consideration that does affect their duration.

**Line-by-line comments**

*l. 55: is the development of CYCLOPS actually more rapid than the one of extratropical cyclones? The organization of convection can be at times very slow (>24h).*

The same could be said of conventional TCs (Zehr 1992). They often go through long gestation periods before real intensification sets in. In both conventional TCs and CYCLOPs, the time timescale for real intensification (which depends on the surface drag coefficient and the potential intensity) is quite a bit faster than baroclinic development time scales.

*l. 68: in which sense "baroclinic" processes? Rossby wave breaking and cut-off formation are rather considered as "barotropic" processes (probably also a not-so- appropriate term...). Could the authors be more specific? Otherwise it might sound like the generation of PI could be tied to baroclinic instability, which is not possible (see comment 1).*

We agree with this and in most places have substituted "dynamic" for "baroclinic". Even conventional baroclinic instability can lead to local (not global) conversions of kinetic to potential energy, as happens, for example, near the ends of the lifecycles of easterly waves.

*l 90 "moistened": I guess the authors are here referring to something like "humidified" or "brought closer to saturation", because cooling alters the relative humidity rather than the specific moisture.*

That is correct, and we have changed "moistened" to "humidified", where appropriate.

*l 91-94: two comments on this sentence, asking for further clarification/contextualization: 1) while the PV argument is elegant and correct, it might not immediately be clear to the reader why such a cancellation between upper PV and lower theta contribution is needed in a minimal conceptual model of CYCLOPs; 2) surface cyclones are often observed at the leading edge of a moving cut-off/PV streamer - e.g., to the east of a streamer moving from west to east- so one would not automatically expect a vertical alignment as in Fig. 1a.*

We have added language to make it clear that we are employing an idealization which is seldom exactly true in nature.

*l 97-98: this is an interesting observation: do the authors observe that the scale of the PV anomaly and the scale of the CYCLOP are correlated across their range of case studies?*

This is a good question, but it would take considerable further analysis to quantify this so we leave it to future work.

*l 118: could a legend table be added -maybe in the appendix- to show the units and the explanation of each symbol?*

We feel that we have described the terms fairly well in the appendix.

*l 131-132: please specify that the surface fluxes are able to destroy the upper-level cold low thanks to the crucial effect of deep moist convection - otherwise their effect would not be "felt" outside the boundary layer. While the link to deep convection is more obvious in the Tropics, it is important to remind it for systems occurring away from warm tropical waters.*

We agree and have modified the text in all the relevant places to indicate that the proximate warming is through the agency of deep convection.

*Section 2: this "Motivation section" actually extends only until line 171, while the remaining part is more an intro to Sec. 3 and the authors could consider moving it between line 203 and line 204. Furthermore, I would suggest the authors to add another motivation item besides predictability and ocean interaction, concerning the role that surface flux-driven cyclones will play in the future climate. In a warmer climate with higher SSTs, areas favorable to TC genesis will extend poleward and surface-flux driven cyclones will likely become more frequent and familiar to the extratropics. The approach outlined in this paper allows to objectively diagnose the presence of surface-flux driving in dubious cases, and even break it down in different contributions to PI. It provides a useful framework to talk about such systems and distinguish them from "usual" cyclones, and provides us with useful tools to quantify and study this effect of climate change onto the extratropical circulation.*

We very much agree with both points and in response to the first have moved the indicated paragraphs into section 3, as the reviewer advised.  It reads much better now. While we also agree with the second point, we elected not to discuss this here as our paper is already pretty long.

*l 226: "(geopotential) height" instead of "pressure"? Pressure is constant on the 950hPa iso-surface.*

Yes indeed, this has now been fixed.

*l 227: where is the "cold pool" visible?*

We have replaced the term "cold pool", which has a rather specific meaning, with "cold air under the cutoff low"

*l 233: the wording raises the question: at least it should be shown in some way somewhere, before it can be claimed with "no doubt" (thanks for specifying here the role of deep convection!).*

Wording changed to be more equivocal.

*Fig. 4 (and others): I see the challenge in finding a common unifying theme for the many figures across the manuscript, but nevertheless there are some aspects that could be improved:*

- *Legend labels, titles, axis labels should be made more visible*

- *Given that only selected contours are shown, consider moving the color map to discrete steps rather than to continuous, so that readability can be improved.*

- *Consider adding 2PVU contour to 400hPa geopotential plot, as PV is part of the paradigm outlined in Fig. 1 and could indicate an even better match with surface features.*

- *Consider using fewer contours lines.*

We have redrawn almost all the map figures to make the labels, titles and legends more readable. We reduced the area covered in Figure 15, also making that figure more legible. Adding the 2PVU contour to the 400 hPa maps is a good idea, but the processing to accomplish it would have taken quite a long time given the software we were using.

*l 247: 750hPa*

Fixed.

*l 274-276, and also 11.600-608: tropical transition is sub-optimal as a TC genesis mechanism, and results in weaker TCs (McTaggart-Cowan et al. 2015, Sec.5), that might even go unnamed such as the one discussed by Bentley and Metz (2016) -which bears resemblance to the case discussed here in Sec. 3.5. So, the capability or not of the cyclone to survive after the transition depends on the changing large-scale environment around each storm, and should not be regarded as a condition to define or not the presence or the quality of tropical transition.*

We added text emphasizing that there is a gray area between tropical transition and what we are calling CYCLOP development.

*ll. 312-313: couldn't it be because the upper-level low has also weakened (see comparison between Fig. 8a,b)? If this is the effect of deep convection, it would be nice to show it or to specify the hypothesized chain of processes.*

We added text to the effect that the reduction of modified PI in this case was likely due to surface heating, redistributed aloft by convection. In other cases, dynamically induced warming of the column when the parent cutoff cyclone is reabsorbed into the westerly current could be important as well.

*Fig. 8b: label b) is missing*

Fixed.

*Sec. 3.3 is very fascinating, both figure and discussion!*

Thanks for this comment.

*Fig. 15: the description at lines 404-412 would greatly profit of labels or symbols added to the figures so that the track of the parent upper-level low can be followed. In addition, the authors could consider whether the chosen domain could be made smaller and some time steps skipped.*

We added markers of the position of the upper-level low and made the domain smaller, but we retained all the original times.

*Fig. 16: the location of development with the upper-level cut-off low to the south-west of the surface low reminds me of the favorable TC-trough interaction by Fischer et al. (2019). Do also the other cases tend to develop with the upper-level cyclone to the SW of the low?*

This is a good question but we do not feel we have enough cases to answer it definitively.

*II. 448-453: please refer to specific sub-plots that best depict the anticyclonic wave breaking (by the way, shouldn't it be cyclonic, and why does it matter?), the presence of the cold pool, and the deep trough.*

We recognize this as anticyclonic wave breaking and are now more specific about when and where this takes place.

*II. 515-519 and Fig. 22: could the authors explain what the additional discussion of the Kona low depicted in Fig. 22 adds to the -already quite heavy- manuscript?*

We feel that the second case is more unambiguously a CYCLOP, but it was from the pre-satellite era and so does not have any imagery. We added text to this effect.

*II. 541-545: the humidifying effect of the middle troposphere is actually consistent with the CYCLOP paradigm, as well as the effect of the Rossby wave breaking in bring the initial disturbance to trigger the storm. If it were not for the pre-existing PI, Zorbas would have been classified as a "classic" CYCLOP: a surface-flux driven storm that owes it existence to the preconditioning of an upper-level trough. I do not see why this should not be identified as a CYCLOP, although it is certainly interesting to point out the pre-existing PI in the Mediterranean, a condition that is -I guess- relatively common in September and becoming more and more common with anthropogenic global warming.*

By our definition, if there is sufficient pre-existing PI and little modification of the PI by the synoptic scale dynamics, then we define the event as a tropical transition. This is the case here. We agree that with global warming, events like these are likely to become more common.

*ll. 627-628: An additional point of ambiguity is the comparison between "pure" tropical transition and "pure" CYCLOPs, because tropical transition is a process, while CYCLOPS are weather systems.*

We have re-worked this paragraph and eliminated Figure 30.

*Fig. 24: would it be possible to show other time steps of PI to better depict the connection between PI and the upper-level flow?*

Yes, it would be possible, but we feel the single figure is adequate to demonstrate the existence of large PI in the pre-storm state.

*ll. 666-667: could the authors discuss a bit more the appropriateness of this assumption in the extratropics or in the dry-convecting environments of polar lows?*

We have added a footnote here to clarify this.

*ll. 673-676: it is not immediately clear which temperature is referred to here by saying "under the cut-off cyclone": the one at the tropopause, or the one at low-levels?*

We have re-worded this sentence for clarity.

**Reviewer 2**

We thank the reviewer for this very helpful review. Here are our responses to the major and minor comments. Quotations from this review are italicized in what follows.

**Major comments**

1. *As the authors mention that cyclops closely resemble the strongly baroclinic cases of tropical transition (Lines 70-71), many tropical transition cases appear to be associated with baroclinic and asymmetric dynamics (e.g., Davis and Bosart 2004, Hulme and Martine 2009, Bentley et al. 2017). On the other hand, the schematics of the three stages in Fig. 1 may give an impression that the dynamics are highly axisymmetric. If the authors consider that the concept of cyclops is also applicable to the cyclones that develop through different (e.g., less symmetric) pathways to the final stage (Fig. 1c), I recommend describing the dynamics of tropical transition in more detail and discussing its relationship with Fig. 1, which may help readers understand the applicability of the cyclops concept to real cases.*

We agree that Figure 1 may inadvertently lead the reader to think that the development is axisymmetric. Rather than re-do the figure, we have added text to the effect that Figure 1 should be thought of as an idealization and that real developments are seldom close to axisymmetric.

2. *The authors propose the potential intensity modified by an upper cold low. I agree that incorporating the influence of upper cold air improves potential intensity. My concern is to what extent the assumption of the moist adiabatic lapse rate is valid to determine s\* (the moist entropy of the boundary layer) below the upper cold low. At least, the lapse rate outside the upper cold low is considered to be the value between moist adiabatic and dry adiabatic in general. The authors should elaborate on this assumption in more detail. Are there any thermodynamical reason or observational evidence that support this assumption? Or, do the authors think that this assumption is ad hoc and plan to improve it in future work?*

It has been our experience that wherever deep convection occurs over water, the lapse rates are close to moist adiabatic. Although we did not examine that assumption closely here, there is a fairly extensive literature on the subject, and we have added a footnote to the Appendix on the consequences of that assumption not applying.

**Minor comments**

1. *Line 73: Replace "David" with "Davis."*

Done.

2. *Footnote 2: Giving a reference to "effective static stability" would be informative.*

We now provide a reference.

3. *Lines 147-148: Giving a reference to strongly frontogenetical process near eye-walls would be informative.*

Done.

4. *Figure 4 and other figures: Replace "GMT" with "UTC."*

We have replaced GMT with UTC everywhere except in the headers of some of the figures, where doing so would have been time consuming. We agree that UTC is more correct, but using GMT at least does not cause any ambiguity.

5. *Figure 8: Add label (b) to the corresponding panel.*

Done.

6. *Lines 374-375: Although polar lows are generally accompanied by large surface sensible heat flux compared to tropical cyclones, is it accurate to say that almost all the surface heat flux is in the form of sensible? This sounds as if the ratio of sensible heat flux exceeds 90 %.*

We replaced "almost all" by "most". At temperatures close to freezing, typical of polar low surface environments, most of the surface fluxes are sensible.

7. *Figure 15: I think that the number of panels can be reduced. In addition, the quality of the figure looks low.*

Although we did not reduce the number of panels in Figure 15, we did reduce the domain size and the legibility of the titles and labels, so the figure is now easier to view.

8. *Figure 22: I recommend switching between panels (a) and (b) as in Fig. 21; i.e., placing the 400 hPa field in the top panel and the 950 hPa field in the middle panel.*

Good idea, and we have done so.

9. *Lines 580-582: Where is the principal upper-level cutoff cyclone in Fig. 29?*

It is hard to say, given the complexity of the flow in this case. We just eliminated the word "principal".

Reference

Zehr, R. M., 1992: *Tropical cyclogenesis in the western North Pacific*, https://repository.library.noaa.gov/view/noaa/13116/noaa_13116_DS1.pdf.

---

## Author Response (AR2)

Responses to the Reviewers of the Revised Manuscript

**CYCLOPs: A Unified Framework for Surface Flux-Driven Cyclones Outside the Tropics**

We thank the reviewers for their second round of reviews and here respond to their suggested minor revisions.

**Response to Reviewer 1:**

First, we apologize for not explicitly stating line numbers that we revised; we thought that the revisions would be clear in the tracked-changes version of the paper. Also, while we stated that we had included a new reference to the point about the simultaneous operation of both advection and surface enthalpy fluxes, we in fact forgot to include that reference. The reviewer correctly surmised that the reference is Fantini (1990). It is now included in the present version.

As to the reviewer's Point 1: The reviewer argues that cyclone amplification cannot occur simultaneously by surface enthalpy fluxes and advection, but this clearly does happen in the model described by Fantini (see, e.g., his Figure 15). The model itself is not highly idealized (it is a nonlinear, nonhydrostatic, convection-permitting model) but it is run in idealized environments, which permits him to, for example, unambiguously specify the initial air-sea thermodynamic disequilibrium. Later work, summarized to some extent in the Wernli-Gray 2024 paper, does not very clearly distinguish whether any added amplification by surface enthalpy fluxes occurs simultaneously with or before or after the baroclinic phase.

The point of contention here is not central to our current paper, but we have slightly revised the paragraph on lines 64-72 to try to compromise between the reviewer's view and ours. If the reviewer is so inclined, we would be happy to engage him or her in an offline discussion of this point if he or she is so inclined.

On Point 2: We share the reviewer's view that the there is a gray zone between tropical transition and CYCLOP development, and have modified the language near line 626 to reflect this. We would point out that if there is no antecedent potential intensity whatsoever, then surface flux-driven cyclogenesis must be the result of local PI generation and thus could be considered a "pure" CYCLOP. Likewise, if the enhancement of antecedent PI by the upper-level trough is trivial, as seems to have been the case, for example, in Hurricane Daniel, then the development is much closer to pure tropical transition. In between these two limiting cases is probably a continuum. In our view, the existence of such a continuum in no way negates the usefulness of the CYCLOP concept, anymore than the continuous spectrum of the strengths of upper-level systems that trigger tropical transition lessens the usefulness of that concept.

On the reviewer's detailed comment: We feel that Figure 1 benefits from its comparative simplicity and think that adding in a potential pre-existing surface low (or high for that matter), while it would add generality to this conceptual figure, would also add complexity that would make the sequence potentially more difficult to follow, so we elect not to modify that figure or the existing discussion around it. About lines 243-245: We have added a suitable reference.

**Response to Reviewer 2:**

We appreciate that we caused some confusion in switching from a modification of the potential intensity based on the upper-level geopotential perturbation to one based on the near-surface geopotential perturbation. The first author takes full responsibility for this change. The original equation stemmed from his earlier work on medicanes (Emanuel, 2005) and sought to answer a different question: Given some climatological mean state, how would potential intensity be affected by the insertion of an upper-level, cold-core cutoff cyclone? But here the objective is very different. The upper-level, synoptic-scale cyclone is likely to be well represented in the ERA-5 reanalyses, so no modification of the potential intensity calculated using the reanalysis data near the upper cyclone center is necessary. On the other hand, once a warm core, surface flux-driven cyclone begins to develop under the cutoff low aloft, the warming of the column modifies the pre-existing potential intensity under that upper-level cyclone. We want to know what the potential intensity was *before* the warm core cyclone developed. That is what the new correction accomplishes.

The same exact problem occurs for normal tropical cyclones. Potential intensity calculated from operational analyses usually shows a diminishment of PI near the TC center. In the case of a very strong TC, this may cause the PI to approach zero near its center. For this reason, when we need PI as an input to a TC model, we usually use the PI several days before the analysis in question. Given the rapid evolution and local nature of PI in the case of a CYCLOP, we do not have this luxury and thus use equation (1) of the paper to correct for the presence of a CYCLOP in the calculation of PI.

Just before equation (1) in the new version of the paper, we state that "*One practical challenge is calculating the potential intensity. This should be calculated using the temperatures of the sea surface and the free troposphere under the PV anomaly aloft, but before the troposphere has appreciably warmed from surface fluxes. In practice, because the warming occurs either as the PV anomaly develops over water, or as it moves over water from land, we have no access to the sounding of the free troposphere before it has warmed up. The true potential intensity for a TC developing within the cold column under the PV anomaly is hence impossible to obtain. Nevertheless, we can estimate how cold the troposphere was before surface fluxes warmed it by using the surface pressure perturbation as a proxy for the CYCLOPs-induced warming, and assuming that the troposphere has an approximately moist adiabatic temperature profile. This is derived in the Appendix*". The first paragraph of the Appendix itself reads "*CYCLOPs form when dynamical processes create synoptic-scale cold columns marked by cutoff cyclones in the upper troposphere. The objective here is to calculate tropical cyclone potential intensity in these cold columns. The problem is that the CYCLOPs themselves, and surface heat fluxes in general, warm the columns, sometimes rapidly, diminishing the potential intensity. We want to know what the potential intensity was before this warming occurs. Here we develop a modified potential intensity using the surface pressure depression as proxy for the column heating.*" We hope this is now sufficiently clear.

As to this reviewer's minor comments:   Footnote 2 references a research paper by Emanuel et al., not the Emanuel textbook. Both were published in 1994, thus the confusion. We have modified the language around line 744 (now 748) to avoid the confusion noted by the reviewer.

---

## Author Response (AR3)

Response to Editor on Second Revision of

**CYCLOPs: A Unified Framework for Surface Flux-Driven Cyclones Outside the Tropics**

We have basically implemented the editor's suggestion for altering footnote two, which now reads

*Some would regard latent heating as an additional energy source, but condensation through a deep layer is present in most cyclones, being strongly tied to vertical motion, and one could argue that it should not be regarded as an external heat source but rather as a modification of the static stability (Emanuel et al., 1994). On the other hand, large areas of cyclones are not water saturated and various processes affect the availability of moisture in such regions, and thus, ultimately, the amount of latent heat release (e.g. Winschall et al., 2014). Moreover, latent heat release, when it is strong enough, can lead to the phenomenon of diabatic Rossby waves (Boettcher and Wernli, 2013; Kohl and O'Gorman, 2022; Parker and Thorpe, 1995; Wernli et al., 2002) and here the partitioning between advection and latent heating is important.*

This is close to, but not identical with the editor's suggestion. We also addressed the editor's minor point by eliminating two sentences that were no longer relevant to the revised version of the figure and paper.